# Principled Synthetic Data Enables the First Scaling Laws for LLMs in Recommendation

**Benyu Zhang** [* 1]   **Qiang Zhang** [* 1]   **Jianpeng Cheng** [1]   **Hong-You Chen** [1]   **Qifei Wang** [1]   **Wei Sun** [1]   **Shen Li** [1]
**Jia Li** [1]   **Jiahao Wu** [1]   **Xiangjun Fan** [1]   **Hong Yan** [1]

## Abstract

Large Language Models (LLMs) represent a promising frontier for recommender systems, yet their development has been impeded by the absence of predictable scaling laws, which are crucial for guiding research and optimizing resource allocation. We hypothesize that this may be attributed to the inherent noise, bias, and incompleteness of raw user interaction data in prior continual pre-training (CPT) efforts. This paper introduces a novel, layered framework for generating high-quality synthetic data that circumvents such issues by creating a curated, pedagogical curriculum for the LLM. We provide powerful, direct evidence for the utility of our curriculum by showing that standard sequential models trained on our principled synthetic data significantly outperform ($+130\%$ on recall@100 for SasRec) models trained on real data in downstream ranking tasks, demonstrating its superiority for learning generalizable user preference patterns. Building on this, we empirically demonstrate, for the first time, robust power-law scaling for an LLM that is continually pre-trained on our high-quality, recommendation-specific data. Our experiments reveal consistent and predictable perplexity reduction across multiple synthetic data modalities. These findings establish a foundational methodology for reliable scaling LLM capabilities in the recommendation domain, thereby shifting the research focus from mitigating data deficiencies to leveraging high-quality, structured information.

---
[*]Equal contribution   [1]MRS, Meta.   Correspondence to:   Benyu Zhang <byzhang@meta.com>, Qiang Zhang <qiangzhang@meta.com>.

*Proceedings of the $43^{rd}$ International Conference on Machine Learning*, Seoul, South Korea. PMLR 306, 2026. Copyright 2026 by the author(s).

## 1. Introduction

Large Language Models (LLMs) represent a promising frontier for recommender systems, offering advanced sequence modeling capabilities and extensive world knowledge that can transcend the limitations of traditional embedding-table architectures (Kang & McAuley, 2018; Zhai et al., 2024). By representing users, items, and interactions within a unified linguistic space, LLMs can leverage textual descriptions to generate more sophisticated, content-aware, and explainable recommendations (Rajput et al., 2023; Chen et al., 2024; He et al., 2025). The development of such systems at scale, however, requires the establishment of predictable scaling laws—indispensable instruments for navigating the substantial investments in data, computation, and engineering that modern LLM development demands (Hoffmann et al., 2022). Yet, despite their established significance in NLP, *no robust scaling laws have been established for the continual pre-training (CPT) of LLMs within the recommendation domain* (Zhou et al., 2025a; He et al., 2025), compelling researchers to depend on costly trial-and-error methodologies.

Prior work on scaling LLMs for recommendation has taken either model-centric (Yan et al., 2025), distillation-centric (Lai et al., 2025), or system-centric approaches, while still relying on raw user interaction logs as the primary data source. Such logs are characterized by systemic flaws: noise, sparsity, and—most critically—pervasive biases including position bias, popularity bias, and exposure bias (Chen et al., 2021; Dooley et al., 2023). Recent work has shown that scaling laws only manifest for data of sufficient quality and diversity (Yang et al., 2024; Allen-Zhu & Li, 2024a). For example, the PLUM framework (He et al., 2025) encountered suboptimal scaling behavior where a 3B parameter model failed to consistently outperform its 900M counterpart—a clear symptom of training on information-poor, structurally biased data.

Our central insight is that prior scaling failures are not primarily algorithmic, but rather stem from a deficient data paradigm. Rather than designing models that are robust to data imperfections, we shift focus to constructing high-quality, "pedagogical" data that is engineered to teach the

model the principles of recommendation in a structured manner. We introduce a novel, layered synthetic data framework that systematically decouples true user preference signals from system-induced artifacts. Layer 1 establishes foundational knowledge through item-text alignment and collaborative filtering data, while Layer 2 generates position-debiased user interaction histories via graph-based random walks that have no intrinsic concept of position or presentation order.

We provide powerful, direct evidence for the utility of this framework. First, standard sequential recommendation models (GRU4Rec (Hidasi et al., 2016), NARM (Li et al., 2017), STAMP (Gao et al., 2025), and SASRec (Kang & McAuley, 2018)) trained on our synthetic data achieve **significantly higher Recall@K** than models trained on real data across all cutoff points, demonstrating that our data captures more generalizable user preference patterns. Second, we empirically demonstrate, **for the first time**, robust power-law scaling for LLMs continually pre-trained on this data. Across model scales from 0.6B to 8B parameters trained on 163B tokens, we observe consistent scaling behavior characterized by the law $L(D) = L_\infty + A \cdot D^{-\alpha}$. Third, ablation studies reveal an asymmetric synergy between data layers: including collaborative filtering data alongside user interaction histories reduces asymptotic UIH loss by 31% ($L_\infty = 0.66$ vs. $0.95$), while the reverse transfer does not hold—demonstrating that our layered curriculum provides complementary, non-redundant learning signals.

Our quantitative analysis reveals a clear hierarchy of learning efficiency across data modalities. User Interaction History (UIH) data exhibits exceptional scaling exponents ($\alpha \approx 0.45$–$0.59$), indicating the model continues to benefit substantially from additional training tokens. Collaborative Filtering (CF) data shows strong scaling ($\alpha \approx 0.35$), followed by Item-Text alignment ($\alpha \approx 0.15$). In addition, our experiment has shown introducing CF data helps the knowledge acquisition of UIH data. These findings establish a foundational methodology for reliable, predictable LLM development in the recommendation domain, providing practitioners with the first quantitative roadmap for resource allocation and performance forecasting.

This paper makes the following contributions:

- **A layered synthetic data framework** that transforms noisy, biased user interaction logs into a high-quality curriculum. Layer 1 grounds semantic and collaborative knowledge through item-text alignment and association-rule data; Layer 2 generates position-debiased user interaction histories via graph-based random walks that eliminate positional and presentation-order artifacts.

- **Direct empirical validation of data utility**: We demonstrate that standard sequential models trained

exclusively on our synthetic data outperform models trained on real data in downstream ranking tasks (Recall@10, @100, @1000), providing strong evidence that our curriculum captures more generalizable user preference patterns than raw interaction logs.

- **The first scaling laws for LLMs in recommendation**: Across model scales from 0.6B to 8B parameters trained on 163B tokens, we establish robust power-law scaling across seven evaluation domains. User interaction history exhibits the strongest scaling ($\alpha \approx 0.45$–$0.59$), followed by collaborative filtering ($\alpha \approx 0.35$) and item-text alignment ($\alpha \approx 0.15$), revealing a hierarchy of learning efficiency that provides a quantitative roadmap for data curation and resource allocation.

- **Discovery of asymmetric cross-domain transfer**: Ablation studies reveal that collaborative filtering data provides complementary signals that significantly improve user behavior modeling ($L_\infty$ reduced from 0.95 to 0.66, a 31% improvement), while the reverse transfer does not hold. This validates the pedagogical design of our layered curriculum.

## 2. Related Work

**Scaling Laws in Recommender Systems.** Scaling laws (Kaplan et al., 2020; Hoffmann et al., 2022) characterize how model performance improves with parameter count ($N$), dataset sample size ($D$), and compute in FLOPS ($C$): $L(N, D) = AN^{-\alpha} + BD^{-\beta} + E$. However, no such predictable scaling has yet to be established for LLM based recommendation models. Recent approaches pursue different strategies: Large User Models (LUM) (Yan et al., 2025) introduce "next-condition-item prediction" to capture contextual preferences; SUAN (Lai et al., 2025) uses knowledge distillation from accurate teacher models; PLUM (He et al., 2025) adapts LLMs via CPT on user-item interactions but does not systematically study scaling laws; and OneRec (Zhou et al., 2025a;b) studies scaling in model size and test-time compute.

These works are either model-centric, system-centric, or distillation-centric. As a fundamentally *data-centric* solution, our work is orthogonal and complementary—We posit that even advanced architectures struggle to exhibit power-law scaling when trained on pathologically flawed data. By solving the data quality problem first, our framework provides the foundation for predictable scaling.

**Synthetic Data for Recommendation.** Synthetic data has been used to understand LLM training dynamics (Yang et al., 2024; Allen-Zhu & Li, 2024a; Hron et al., 2024) and in industry LLM training (Yang et al., 2025a). In recommender systems, synthetic data traditionally focuses on augmen-

tation (Qi et al., 2020) and privacy preservation (Adouani & Dagdia, 2025) using GANs (Bharadhwaj et al., 2018) and VAEs (Adouani & Dagdia, 2025) that replicate source data distributions. Our framework represents a conceptual departure along three dimensions: (i) *GAN/VAE-based augmentation* replicates the source distribution, inheriting its biases; our framework instead *purifies* the signal by generating data from a debiased CF graph. (ii) *Sequence augmentation* methods (e.g., crop, mask, reorder (Qi et al., 2020)) produce perturbed copies of existing sequences; our Layer 2 generates entirely new sequences via graph-based random walks. (iii) *LLM-based generation* operates at the sample level; our framework operates at the *curriculum level*, designing structured training stages that systematically build recommendation capabilities (Chen et al., 2025a; Soviany et al., 2022).

**Debiasing Recommendation Data.** The debiasing literature encompasses inverse propensity scoring (Cardoso et al., 2022; Agarwal et al., 2019; Wang et al., 2018), causal inference (Pan et al., 2025; Li et al., 2023), and adversarial training (Yang et al., 2025c; Feng et al., 2019). These approaches design complex models robust to bias. Aligned with data-centric AI (Zha et al., 2023), we take a more direct approach: instead of building bias-robust models, we engineer position-debiased data, enabling standard LLM architectures without bespoke debiasing modifications.

**Graph-Based Methods for Recommendation.** GLTA (Yang et al., 2025b) integrates collaborative filtering with LLMs via graph-language token alignment, treating user-item interactions as graph modalities. While this is an interesting direction, its scalability to catalogs of hundreds of millions of items remains unclear, and it may lack the reasoning capabilities that LLMs offer. S-Walk (Choi et al., 2022) uses random walks to model inter-session relationships for session-based recommendation; its explicit inter-session modeling could complement our Node2Vec-based sequence generation in Layer 2. Our framework differs fundamentally from both: rather than augmenting model architectures, we construct a pedagogical data curriculum that enables standard LLM training to exhibit predictable scaling.

## 3. The Data Quality Bottleneck in Recommendation CPT

Adapting LLMs for recommendation requires bridging the domain gap between general-purpose language knowledge and idiosyncratic user behavior patterns. The quality of adaptation data is paramount, yet raw user interaction logs can be flawed training sources.

**Systemic Biases in User Logs.** Raw logs are not pure reflections of user preference but distorted records influenced by the systems that generated them (Chen et al., 2021; Dooley et al., 2023). Table 1 categorizes these pathologies: *incompleteness* (fragmented cross-platform profiles), *noise* (accidental clicks, contradictory ratings), *position bias* (CTR drops 50%+ from rank 1 to 5 (Bito et al., 2025)), *popularity bias* (top 1% items receive 80% of interactions (Braun et al., 2023)), and *exposure bias* (users only interact with shown items (Mansoury et al., 2022)). LLMs readily internalize these artifacts, mistaking them for genuine user intent.

*Table 1.* Taxonomy of systemic biases in user interaction logs. These artifacts are strong, consistent patterns that LLMs readily learn, mistaking them for genuine user intent.

| Bias Type | Description | Example |
|---|---|---|
| **Incomplete & Sparsity** | Users interact across multiple platforms; profiles and attributes are partial | User buys camera on Amazon, accessories on B&H—neither platform sees full intent |
| **Noise** | Clicks may be accidental; ratings contradict reviews | 5-star rating with review: "Arrived broken, returning it" |
| **Position Bias** | Users favor top-ranked items regardless of relevance (Bito et al., 2025) | CTR drops 50%+ from position 1 to 5, even for equally relevant items |
| **Popularity Bias** | Popular items get more exposure, creating feedback loops (Braun et al., 2023) | Top 1% of items receive 80% of interactions; long-tail items never surface |
| **Exposure Bias** | Users can only interact with shown items (Mansoury et al., 2022) | User loves jazz but system only recommends pop—logs show zero jazz preference |

**Sub-Scaling from Low-Quality Data.** The PLUM (He et al., 2025) framework involves a CPT phase that utilizes domain-specific data, including raw user activity sequences, to adapt a pre-trained LLM to recommendation tasks. However, empirical results indicated that this approach faced significant scaling challenges. Specifically, the performance of the MoE-3B model was constrained by insufficient training data, ultimately preventing it outperforming the substantially smaller MoE-900M model. This aligns with recent findings that high data density and redundancy—characteristics of raw logs with popularity-driven feedback loops—cause performance gains to diminish (Chen et al., 2025b; Tirumala et al., 2023). When an LLM is trained on biased logs, it codifies and amplifies biases more effectively than simpler models. Deploying this LLM creates a vicious cycle where skewed recommendations generate more contaminated logs, degrading quality over subsequent rounds of CPT. The only escape is to fundamentally alter the training data itself.

## 4. A Layered Framework for High-Fidelity Synthetic Data

Rather than mitigating biases in existing flawed data, we construct an entirely new, high-fidelity dataset designed as a structured curriculum. This aligns with findings that synthetic continued pretraining with diverse, grounded data outperforms repetition of source documents (Yang et al., 2024; Allen-Zhu & Li, 2024a). Our framework organizes data into two layers that progress from foundational concepts to integrated behavioral patterns.

### 4.1. Layer 1: Semantic and Collaborative Grounding

**Item-Text Alignment.** This data establishes semantic links between item identifiers (semantic tokens) and textual descriptions, enabling content-based reasoning beyond ID co-occurrence.

**Collaborative Filtering (CF).** Association rules are mined from raw user interaction logs via co-occurrence statistics across user sessions. The resulting item-item relationships are translated into templated natural language statements that make collaborative patterns explicit:

> **When a user interacts with item <RECTO-KEN> REC3078 ... </RECTOKEN>, there is a 4.9% probability they also interact with <REC-TOKEN> REC3078 ... </RECTOKEN> (lift: 652.45)**

### 4.2. Layer 2: Position-Debiased User Behavior Simulation

**Synthetic User Interaction Histories (UIH).** We generate clean, privacy-preserving interaction sequences using the CF relationships from Layer 1. A graph is constructed with items as nodes and CF co-occurrence weights as edges. User journeys are simulated via Node2Vec (Grover & Leskovec, 2016) 2nd-order biased random walks, where the transition probability $\pi_{v,x} = \alpha_{p,q}(t,x) \cdot w_{v,x}$ depends on both current node $v$ and previous node $t$. The return parameter $p$ controls backtracking likelihood to the previous node $t$, and the in-out parameter $q$ controls exploration ($q > 1$) vs. exploitation ($q < 1$). We performed hyperparameter sweeps over $p$, $q$, walk length, and the number of walks per node, and found the scaling law to be robust across configurations (full sweep in Section F).

This layered approach acts as a data purification process. Layer 1 extracts core co-occurrence signals by averaging over sessions, reducing susceptibility to position bias. Layer 2 generates sequences from this purified signal—the random walk process has no concept of "rank" or "presentation order," ensuring that UIH data is free from positional and

presentation-order artifacts. We note that while this process eliminates position and temporal-ordering biases by construction, popularity bias encoded in graph edge weights may persist; we provide a quantitative bias audit in Section E. The evaluation test sets and their definitions (CF Both Seen, CF One Unseen, CF Both Unseen) are detailed in Section A. As an example:

> **A user interacted with: <RECTOKEN> REC870 REC5932 ... </RECTOKEN>, ..., <RECTO-KEN> REC7402 REC1581 ... </RECTOKEN>**

## 5. Empirical Validation of Synthetic Data Quality

The central thesis of this work is that high-quality synthetic data is the key to unlocking scaling laws. To empirically substantiate this claim, we evaluate the generated data across three dimensions: fidelity and utility. By design, our data generation process offers inherent privacy benefits.

### 5.1. Statistical Fidelity and Bias Analysis

Fidelity measures how closely the synthetic data mirrors the statistical properties of the original data. We compared the distributions of key characteristics, such as item popularity and sequence length, between the generated Synthetic UIH and the original Merrec user logs. To quantify the debiasing properties of our framework, we conducted a comprehensive hyperparameter sweep (14 configurations, 100K sequences each; see Section E). Key findings: (i) **Item exposure inequality**: Synthetic UIH achieves Item Gini coefficients of 0.635–0.732 (production config: 0.639), indicating moderate and healthy item concentration—substantially more uniform than real logs where top 1% of items receive 80% of interactions. (ii) **Long-tail coverage**: Synthetic UIH covers 174K–189K unique items per 100K sequences and 88.6–89.5% of the semantic token vocabulary. (iii) **Position bias removal**: The random walk has no concept of "rank" or "presentation order"—each step is determined solely by graph structure and $(p, q)$ parameters, ensuring zero positional artifacts by construction. We emphasize that popularity bias encoded in graph edge weights is *not* fully eliminated; the framework targets positional and presentation-order biases specifically.

### 5.2. Empirical Validation of Data Utility for Downstream Ranking

To assess the practical utility of our synthetic data for the core recommendation task of ranking, we conduct a "Train on Synthetic, Test on Real" (TSTR) evaluation and compare it against a "Train on Real, Test on Real" (TRTR) base-

line. To ensure a fair comparison, this evaluation uses a constrained methodology: the test set, drawn from real user interactions, is filtered to include only items present in the vocabulary of both the synthetic and real training sets.

We trained four standard sequential recommendation models (GRU4Rec (Hidasi et al., 2016), NARM (Li et al., 2017), STAMP (Gao et al., 2025), and SASRec (Kang & McAuley, 2018)) in both the TSTR (Syn→Real) and TRTR (Real→Real) settings. The results, visualized in Figure 1 below, reveal a striking and powerful finding. In all cases, models trained exclusively on our synthetic data (Blue lines) achieve significantly better ranking performance (Recall@K) than models trained on real data (Red lines) across all cutoff points (@10, @100, @1K).

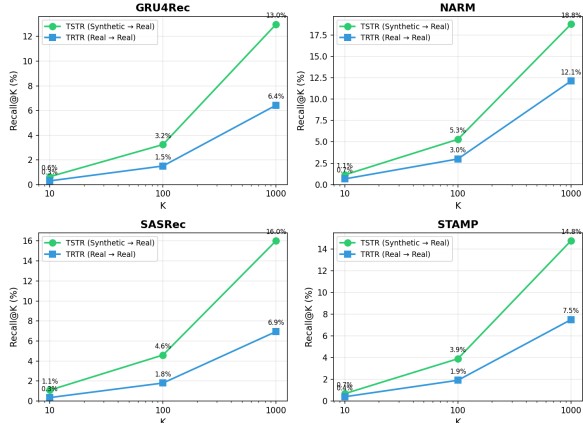

*Figure 1.* TSTR vs TRTR: Recall@K Comparison Across Models. TSTR (Train on Synthetic, Test on Real) consistently outperforms TRTR (Train on Real, Test on Real) across all models (GRU4Rec, NARM, SASRec, STAMP) and K-values. Note: Real UIHs are filtered to the same set of items as Synthetic UIHs.

These results provide strong, direct evidence that our synthetic data is not only a high-fidelity substitute but is in fact a superior training resource for teaching the general principles of user preference required for high-quality recommendation. By learning from the purified collaborative filtering signal rather than the noisy specifics of raw interaction logs, models trained on synthetic data acquire more generalizable co-occurrence patterns. This leads to superior performance on ranking tasks, which require an understanding of the broad set of plausible next items rather than just the most likely one.

## 6. Principled Scaling Laws for Recommendation

### 6.1. Experimental Configuration

Having established the superior utility of our synthetic data for downstream tasks, we now turn to our central hypothesis:

*Table 2.* Dataset statistics for this experiment. Our dataset consists of both general domain data (Ben Allal et al., 2024) and recommendation data from Merrec (Li et al., 2025) We mix those two domains in a 1:1 ratio to avoid catastrophic forgetting (Ibrahim et al., 2024) Recommendation data consists of three types synthetic data from two layers as described in Section 4. All models are trained with 163B tokens; the last row estimates the number of repeats each domain would be exposed to after training.

| Domain | General | Item Text | CF | UIH |
|---|---|---|---|---|
| Token | 128B | 2.0B | 7.6B | 2.8B |
| Ratio | 50% | 9% | 30% | 11% |
| Repeats | $< 1$ | 6.2 | 7.3 | 6.1 |

that this high-quality data enables predictable scaling laws for LLMs. To empirically validate this, a series of continual pre-training experiments were conducted with the following settings:

- **Base Model**: We use Qwen3 models (Yang et al., 2025a) at 0.6B, 1.7B, 4B and 8B-parameter scales, which is one of the best open-weight LLM model available now. We intialize from pre-trained checkpoints.

- **Dataset**: Table 2 describes the statistics of our training dataset.
  - **General**: We use three subsets (cosmopedia-v2, fineweb-edu-dedup and python-edu) from the SmolLM-Corpus (Ben Allal et al., 2024) as our general domain data to avoid catastrophic forgetting during continual pretraining.
  - **Recommendation**: We use the the merrec dataset (Li et al., 2025), a publicly available C2C (consumer-to-consumer) e-commerce dataset from HuggingFace with 1.2B interactions from 65.7M items and 2.6M users. We use sparse auto-encoder generated semantic ID to represent the item (Section B).

- **CPT Hyperparameters**: Continual pre-training was performed on B200 clusters using a global batch size of 512 sequences, a context window of 512 tokens, a learning rate that peaks at 1e-4 with a linear warmup of 100 steps and cosine decay schedule. All models would be trained with 163B tokens, which is over 20x (Chinchilla scaling multiplier (Hoffmann et al., 2022)) for 8B models.

- **Evaluation Metric**: Model performance was measured using perplexity on a held-out test set across all domains (general domain, item-text, CF both seen, CF one unseen, CF both unseen, UIH OOD, and UIH full graph). The details are presented in Section A

*Table 3.* Scaling laws for different domains across different model scales. UIH exhibits strong scaling ($\alpha_{\text{UIH}} = 0.453 - 0.689$), indicating continued improvement with additional training tokens. General domain shows near-saturation ($\alpha \approx 0.025$) as expected for pretrained models. $L_\infty$: the lower the better; $\alpha$: the higher the better.

| Size | General | | Item | | CF | | UIH | |
|---|---|---|---|---|---|---|---|---|
| (# params.) | $L_\infty$ | $\alpha$ | $L_\infty$ | $\alpha$ | $L_\infty$ | $\alpha$ | $L_\infty$ | $\alpha$ |
| 0.6B | 0.96 | **0.03** | 1.07 | **0.17** | 0.33 | 0.28 | 0.64 | 0.45 |
| 1.7B | 0.87 | 0.02 | 0.92 | 0.14 | 0.34 | 0.32 | 0.65 | 0.51 |
| 4B | 0.82 | 0.03 | **0.86** | 0.13 | 0.35 | 0.36 | 0.66 | 0.56 |
| 8B | **0.79** | 0.02 | 0.93 | 0.21 | 0.36 | **0.39** | 0.66 | **0.59** |

## 6.2. Power-Law Scaling Across Data Modalities

Figure 2 shows the evaluation loss curves over the course of training for seven different data modalities. Across all seven conditions—General Domain, Item-Text, CF Both Seen, CF One Unseen, CF Both Unseen, UIH OOD, and UIH Full Graph—empirical data points exhibit a remarkably consistent trend. The fitted scaling law parameters ($\ell(D) = L_\infty + AD^{-\alpha}$ with ideal loss ($L_\infty$) and scaling exponent ($\alpha$)) are summarized in Table 3, with the training loss in Figure 9(a). Qualitative examples are provided in Section D.

This quantitative summary reveals a deeper structure in the model's learning process. The scaling exponent, $\alpha$, can be interpreted as a measure of the pedagogical efficiency of each data modality. In the log-log plot, $-\alpha$ represents the slope of the line; a larger $\alpha$ value corresponds to a steeper downward slope, signifying that the model's perplexity decreases more rapidly for each unit of training data consumed.

Table 3 shows a clear hierarchy of pedagogical efficiency. The Synthetic User Interaction History (UIH) data types exhibit the highest scaling exponents ($\alpha \approx 0.55$), meaning that these are the most training-efficient. This is followed by the Collaborative Filtering (CF) data types ($\alpha \approx 0.35$), which are themselves significantly more efficient than the Item-Text alignment data ($\alpha \approx 0.15$). The General Domain baseline is the least efficient ($\alpha \approx 0.025$). This could be due to the pre-trained checkpoint, which is used as initial state of the continual pretraining (CPT), already contains knowledge of general domain, so CPT on more general-domain data does not add new general domain knowledge effectively.

The 0.6B model has the largest $\alpha$ over all sizes in general domain, as its initial checkpoint has the least amount of general domain knowledge. Larger models generally show larger $\alpha$ for CF and UIH domain, as they are better at acquiring new knowledge; however their $L_\infty$ values are similar, which suggests that even 0.6B parameters sufficient for learning the CF and UIH domains of this dataset.

## 7. Ablation Study and Analysis

### 7.1. Ablation Study: The Synergistic Effect of Layered Data

A core tenet of our methodology is that the different layers of synthetic data are complementary, with each teaching a distinct and necessary facet of the recommendation domain. To validate this hypothesis and quantify the contribution of each layer, we conduct an ablation study with Qwen3-4B model. The study involves running the scaling law analysis on models trained with different combinations of our synthetic data curriculum: (1) UIH data only, (2) CF and UIH data, and (3) the complete mixture of Item-Text, CF, and UIH data. We evaluate loss across multiple test sets and fit scaling laws to characterize asymptotic performance. Table 4 provides the mixture ratio of each domain, with all other settings held constant. Figure 3 shows loss on hold-out evaluation dataset for each domain.

Our hypothesis is that training on the complete, mixed dataset will yield the most efficient scaling, as evidenced by a larger scaling exponent ($\alpha$) compared to the ablated conditions. Such a result would demonstrate a super-additive, or synergistic, effect, proving that the integrated knowledge from all layers enables the model to learn more efficiently than the sum of its parts.

**Asymmetric Transfer Between CF and UIH**. We observe a notable performanceasymmetry in cross-domain transfer. Including CF data alongside UIH significantly improves UIH modeling performance (Figure 4, with CF+UIH achieving the lowest asymptotic perplexity ($L_\infty = 0.66$) on UIH evaluation sets—even outperforming the UIH-only baseline ($L_\infty = 0.95$). This suggests that collaborative filtering signals, which capture user-item affinity patterns, provide complementary information that benefits sequential user behavior modeling. In contrast, the reverse transfer does not hold: training on UIH data alone yields virtually no improvement on CF tasks ($\alpha \approx 0$), indicating that user interaction sequences do not implicitly encode the pairwise preference signals required for collaborative filtering.

**Domain-Specific Data Remains Essential**. Despite the positive transfer from CF to UIH, we find that domain-specific training data remains critical for each task. Models trained without CF data tend to plateau at substantially higher CF loss ($L_\infty = 1.30$), compared to those with CF data ($L_\infty = 0.35$). Similarly, excluding item-text data leads to severe degradation on item-text evaluation ($L_\infty \approx 3$ vs. $L_\infty \approx 1.2$), while models retaining item-text data maintain stable performance throughout training.

**Trade-offs in Multi-Domain Training**. Including item-text data introduces a modest trade-off: while it prevents catastrophic forgetting of item semantics, it slightly increases UIH loss ($L_\infty = 0.76$ for Item+CF+UIH vs. $L_\infty = 0.66$

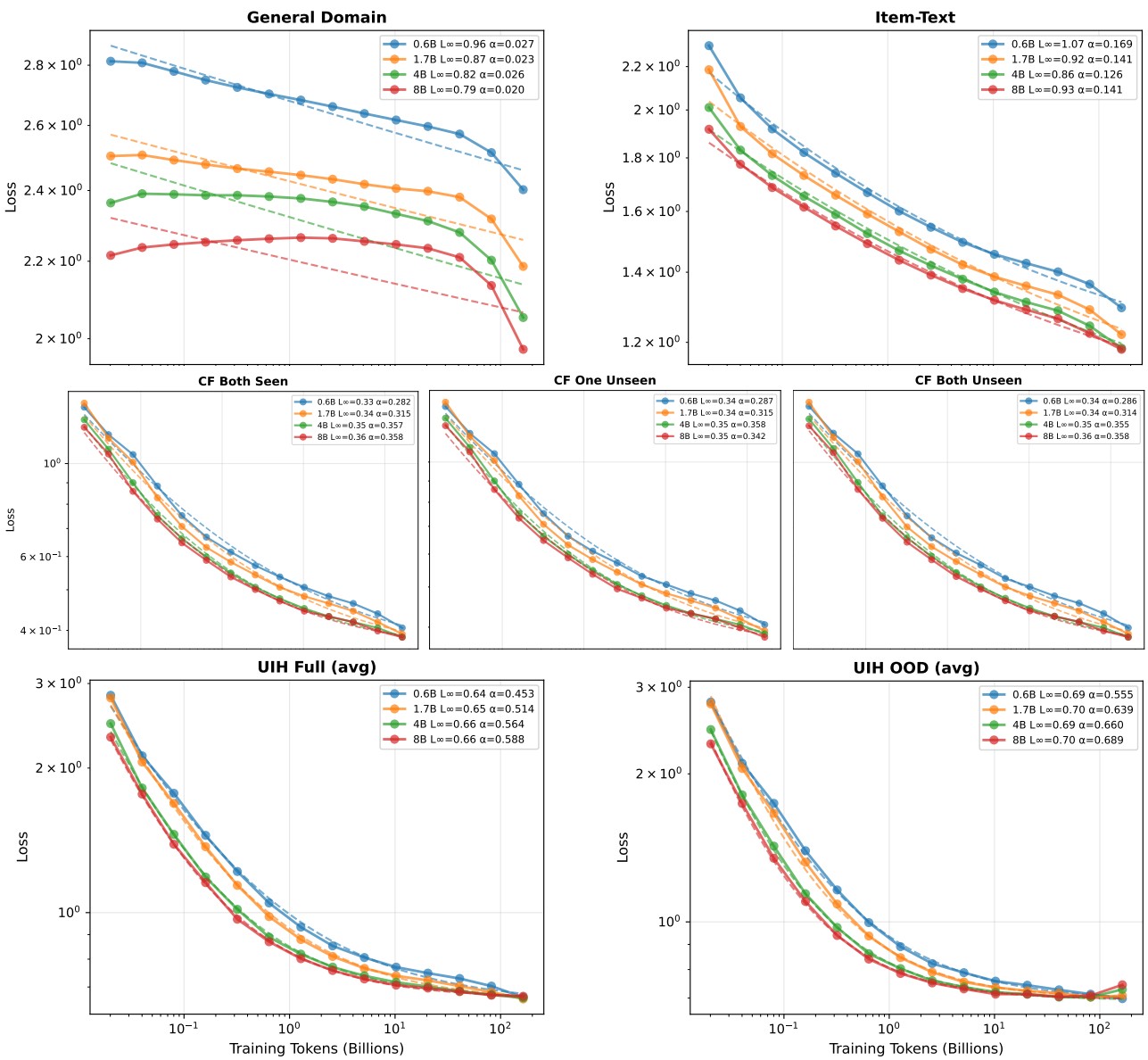

*Figure 2.* Evaluation loss for different domains across different model scales. UIH exhibits strong scaling ($\alpha_{\mathrm{UIH}} = 0.45\text{--}0.59$), indicating continued improvement with additional training tokens. General domain shows near-saturation ($\alpha < 0.1$) as expected for pretrained models. The dashed lines are the fitted scaling law curves and parameters are provided as legend of the curve and Table 3.

*Table 4.* Data mixture ratios of each domain for the experiments. For all experiments we keep 50% for general text and all experiments are trained for 163B tokens.

| Choice & Ratio | Item Text | CF | UIH |
|---|---|---|---|
| Item-text + CF + UIH | 9% | 30% | 11% |
| CF + UIH | 0% | 37% | 13% |
| Item-text UIH | 22.5% | 0% | 27.5% |
| UIH | 0% | 0% | 50% |

for CF+UIH). This suggests that practitioners should consider their downstream priorities when designing data mixtures.

## 7.2. Ablation Study: Data Mixtures

The mixture ratios in Table 2 were determined through a combination of established guidelines and ablation. For the general domain ratio, we use 50% following Ibrahim et al. (2024). For the recommendation domain, we generally follow Muennighoff et al. (2025) and performed ablation on the number of repeats per domain (from 4 to 8), leading to the ratios in Table 2.

A balanced mixture of different data layers is also important to scaling of LLM training. To this end, we sample 42M tokens from 2.7B (or 39K samples from 5.748M samples)

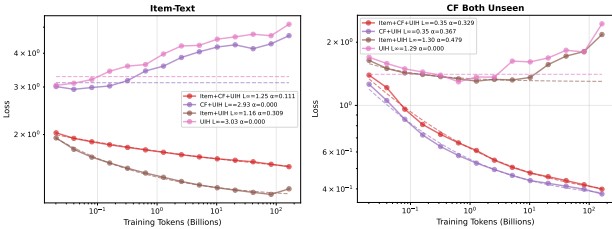

*Figure 3.* Ablation studies on selection of domains for recommendation data. Obviously, excluding a domain leads to degraded performance on the corresponding domain. We omit the figures for CF Both Seen and CF On Unseen here (in Appendix Figure 8), as they are very similar to CF Both Unseen.

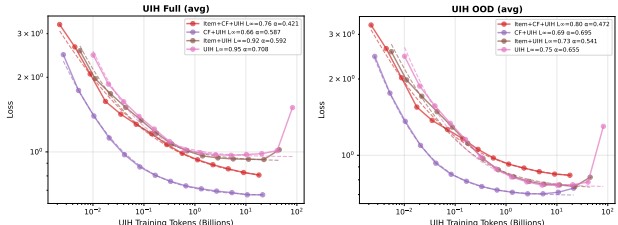

*Figure 4.* We plot the evaluation loss for UIH using number of UIH training tokens as X-axis. This figure indicates including CF data (Item-text + CF + UIH and CF + UIH) enables the model to learn UIH better.

of our UIH data as a reduced UIH data. We then tested the UIH mixture ratio in 0.5%, 1%, 2%, 5% and 15% for this reduced UIH data composition; the mixture ratio of other recommendation domain is normalized accordingly to maintain a sum of 50%. As the UIH mixture ratio on this reduced set increases, the more repeats of the reduced UIH data the model would go through (Column #repeats of Table 5). In this study, we focus on the 4B model and show results in Figure 5.

*Table 5.* Data mixture used in Figure 5. We also provide the number of training tokens and repeats on the UIH data . The first row shows the result using full UIH dataset (baseline), which is the set up in Figure 2; the others are for reduced UIH dataset.

| % | UIH #tokens | #repeats | Item Text % | CF % |
|---|---|---|---|---|
| 11.0% | 17.93B | 6.1 | 9.0% | 30.0% |
| 0.5% | 0.82B | 4.1 | 13.5% | 36.0% |
| 1.0% | 1.63B | 8.2 | 13% | 36.0% |
| 2.0% | 3.26B | 16.3 | 12.0% | 36.0% |
| 5.0% | 8.15B | 40.8 | 12.0% | 33.0% |
| 15.0% | 24.45B | 122.3 | 20.0% | 15.0% |

**Analysis**: Figure 5 indicates that, starting from UIH mixture ratio=2% (about 16 repeats at the end of training according to Table5), the model's perplexity on the hold-out evaluation set starts to increase, which suggests an overfit (according to Figure 9(b), the training loss still decreases monotonically). Increasing ratio further would cause the overfit to happen even earlier: at 15%, it happens at 20B training tokens (about 16 repeats); and at 5%, it happens at 80B training

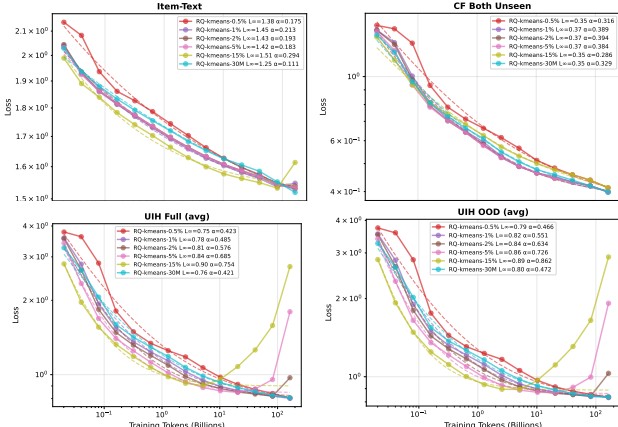

*Figure 5.* Scaling laws on 4B models with different mixture ratio on UIH. Obviously the loss starts to increase when the UIH mixture ratio is too high (reduced UIH data is repeated too many times), which is a sign of overfitting. The higher mixture ratio, this increase happens earlier in training stage. We omitted the figures for CF Both Seen and CF On Unseen here, as they are very similar to CF Both Unseen.

tokens (about 20 repeats).

This result generally aligns with Yang et al. (2024); Muennighoff et al. (2025)'s finding that, given a well constructed synthetic dataset, CPT with four repeats could enable the model to learn the knowledge, and training beyond four repeats gives diminished returns, continuing for 20+ repeats could (Hron et al., 2024) decrease ability to generalize unseen data; performance degrades sharply (Allen-Zhu & Li, 2024b) when repetitions exceed approximately 100× the original dataset size.

We further analyze how model scaling affect this behavior with two UIH mixture ratios at (a) 2% and (b) 15% on the reduced UIH data. Figure 10 in Section C indicates that *when* the perplexity starts to increase is dependent on UIH mixture ratio and independent of model size: the perplexity on evaluation set starts to increase at 20B total training tokens (about 16 repeats on reduced UIH data) when ratio=15%; or at 160B tokens (also about 16 repeats) when ratio=2%, regardless of model scales. However it becomes more significant as the model gets larger.

### 7.3. Scaling Laws Across Model and Data Scale

Our initial results establish that predictable scaling is achievable. The next critical step is to characterize these scaling laws across the two primary dimensions of LLM development—data and model size—to identify compute-optimal training regimes. We conducted compute-optimal scaling analysis across four model sizes (0.6B–8B parameters), all trained on 163.84B tokens with a mixed-domain curriculum (shown in Figure 6). This configuration places models at varying positions relative to Chinchilla-optimal al-

location: the 8B model operates near the optimal 20 tokens-per-parameter ratio, while smaller models are progressively over-trained (0.6B at 454× tokens/param). For general-domain evaluation, where training data is not repeated, we observe classic Chinchilla scaling with a clear Pareto frontier—the 8B model achieves the lowest perplexity at highest compute.

However, recommendation-specific domains exhibit markedly different behavior. Collaborative filtering tasks saturate at model with 4B parameters ( 1.47 PPL), indicating that the pattern complexity is bounded and fully learnable by 4B-parameter models given sufficient repetition. Most notably, user interaction history evaluation on out-of-distribution items reveals *inverse scaling*: the heavily over-trained 0.6B model ( 2.01 PPL) outperforms the near-optimal 8B model (2.10 PPL). We attribute this to the interaction between data repetition and model capacity—larger models more readily memorize repeated behavioral sequences, leading to poor OOD generalization, while the extreme over-training regime of smaller models appears to act as implicit regularization against such memorization.

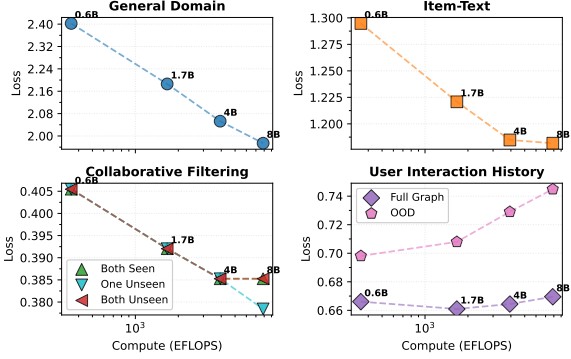

*Figure 6.* Compute-optimal analysis for models from 0.6B to 8B. Our scaling experiments train models from 0.6B to 8B parameters on 163.84B tokens, spanning from heavily over-trained (454 tokens/param for 0.6B) to near Chinchilla-optimal (20 tokens/param for 8B). While general-domain loss follows expected compute-optimal scaling, recommendation tasks—where domain-specific data is repeated 6–7× due to limited unique tokens—exhibit divergent behavior.

## 8. Discussion and Conclusion

This paper addresses a foundational barrier in applying LLMs to recommender systems: the absence of predictable scaling laws. We demonstrate that this barrier stems not from inherent domain limitations but from the pathologically poor quality of raw user interaction data. Our layered synthetic data framework provides a solution by constructing a curated curriculum that decouples true user preferences from system-induced artifacts, enabling the first robust power-law scaling laws for recommendation CPT. Scaling exponents reveal that complex behavioral patterns benefit most from additional data (UIH: $\alpha$=0.45–0.59 vs. Item-Text: $\alpha$=0.15), while asymmetric cross-domain transfer—where CF data accelerates UIH learning by 31%—provides actionable guidance for optimal data mixing.

**Cross-Model and Cross-Dataset Generalization.** We have extended our experiments to a large-scale proprietary social media recommendation dataset with both the **Gemma 3** family (1B, 4B, 12B) and **Qwen3** (0.6B–30B-A3B). Both families exhibit stable power-law scaling with RMSE < 0.035. As shown in Table 6, the hierarchy of learning efficiency ($\alpha$: UIH > Alignment > General) is preserved, and $\alpha$ decreases monotonically with model size across all domains. We also extended the Qwen3-4B token budget from 163B to 1,104B tokens on this dataset, confirming that scaling behavior remains predictable with no saturation. Additional analysis is in Section G.

*Table 6.* Gemma 3 scaling law on an internal dataset (RMSE < 0.035). The hierarchy $\alpha$: UIH > Align. > General is preserved.

| Model | General | | Alignment | | UIH | |
| | $L_\infty$ | $\alpha$ | $L_\infty$ | $\alpha$ | $L_\infty$ | $\alpha$ |
|---|---|---|---|---|---|---|
| 1B | 1.10 | 0.03 | 1.29 | 0.26 | 1.25 | 0.27 |
| 4B | 0.95 | 0.04 | 0.64 | 0.09 | 1.09 | 0.24 |
| 12B | 0.87 | 0.05 | 0.59 | 0.10 | 0.96 | 0.21 |

**Perplexity vs. Downstream Metrics.** While perplexity is our primary scaling metric, several lines of evidence support its practical relevance. The TSTR results (Figure 1) demonstrate that models trained on our synthetic data achieve superior ranking performance on real data, and our extended experiments on the proprietary social media dataset confirmed a strong correlation between perplexity and Recall@K / NDCG for next-item prediction. This is analogous to the Chinchilla scaling law methodology (Hoffmann et al., 2022), which also uses loss-based scaling but became foundational for compute-optimal training.

**Signal Quality vs. Distributional Simplicity.** An important question is whether our improvements come from denoised signals or merely distributionally easier data. Several findings argue for the former: (i) TSTR outperforms TRTR—if synthetic data were simply easier, models should overfit to its distribution; (ii) asymmetric cross-domain transfer (CF→UIH but not UIH→CF) is inconsistent with uniform simplicity; (iii) domain-specific scaling exponents reflect information density, not uniform easiness; (iv) excessive repetition degrades performance (Figure 5), inconsistent with the "easy distribution" hypothesis.

These findings enable practitioners to transition from heuristic-based development to scientific forecasting of computational budgets and data requirements. Future work will extend this analysis to post-training stages and downstream task metrics beyond perplexity.

## Impact Statement

This paper presents work whose goal is to advance the field of Machine Learning, specifically the development of LLMs for recommender systems. We highlight several aspects of broader impact.

**Positive Societal Impacts.** Our synthetic data framework offers inherent privacy benefits: the generated user interaction histories are derived from aggregated item-to-item graphs rather than individual user traces, decoupling training data from real user behavior. Additionally, our methodology addresses key systemic biases (particularly position and presentation-order bias) that plague real-world recommendation systems. By training on position-debiased synthetic curricula with substantially more uniform item exposure (Gini 0.64 vs. >0.95 in real logs), LLMs may produce more equitable recommendations that surface long-tail and niche content.

**Potential Risks and Mitigations.** More effective recommendation systems could, in principle, increase user engagement in ways that raise concerns about digital well-being. However, our work focuses on the *predictability* of model development rather than optimizing for engagement metrics. Furthermore, while our synthetic data generation aims to remove systemic biases, practitioners must validate that downstream applications do not inadvertently introduce new biases through their choice of source data or graph construction. We encourage future work to develop evaluation frameworks that audit both the fidelity and fairness of synthetic recommendation data.

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

*Table 7.* Statistics of the holdout test dataset used in this experiment, which is consisted of four domains: general, item-text, CF and UIH OOD.

| Domain | General | Item Text | CF | UIH |
|--------|---------|-----------|------|-----|
| Token | 102.4M | 102.4M | 243M | 51M |

## A. Hold-out Test Set

Model performance was measured using perplexity on a held-out test set across all domains (general domain, item-text, CF both seen, CF one unseen, CF both unseen, UIH OOD, and UIH full graph). The statistics are presented in Table 7.

- **General domain**: We randomly sample 200K entries (102.4M tokens) for evaluation purpose.

- **Item-text**: item-text pairs for items not in training set. We then sample 200K entries (102.4M tokens) for evaluation purpose.

- **CF both seen**: collaborative filtering edge between two items with items presented in training data but edges are not included in training data. We then sample 200K entries (102.4M tokens) for evaluation purpose.

- **CF one unseen**: CF edge between two items with one item presented in training data and the other items not. Obvious those edges won't present in training data. We then sample 200K entries (102.4M tokens) for evaluation purpose.

- **CF both unseen**: CF edge between two items with neither items nor edges presented in training data. There are about 75k entries (38M tokens) in total.

- **UIH OOD**: UIH sampled only based on edges from CF test set. There are about 48.3K entries (24.7M tokens).

- **UIH full graph**: UIH sampled from both CF train and CF test set. There are about 51.7K entries (26.5M tokens).

## B. Methodology for Semantic Tokenization

This appendix details the methodology for determining the semantic item representation (<RECTOKEN>). As this representation serves as the fundamental vocabulary for our LLM, its quality is paramount. We evaluated three candidate strategies—a pre-trained Sparse Autoencoder (SAE), an in-domain Residual-Quantized VAE (RQ-VAE), and an in-domain RQ-kmeans—and selected SAE based on ablation experiments.

### B.1. Candidate Methodologies

We evaluated two primary approaches for generating discrete item tokens:

- **Approach 1: Pre-trained Sparse Autoencoder (SAE).** This method utilizes a large-scale SAE originally trained on an internal multi-modal dataset comprising video, photo, and text data. The procedure involves passing the item's textual description through the pre-trained embedding model and then into the SAE. The top-k activated features (concepts) from the SAE are selected as the tokens.

  – **Potential Strength**: Leverage of a "foundation" encoder with exposure to massive, diverse multi-modal data.
  – **Critical Risk (Semantic Mismatch)**: The SAE features were learned in a video-centric domain. Applying this vocabulary to e-commerce product text introduces a severe risk of negative transfer, where the "concepts" extracted are mathematically proximal but semantically incongruous for the recommendation task.

- **Approach 2: In-Domain Residual-Quantized VAE (RQ-VAE).** This method involves training a hierarchical RQ-VAE from scratch specifically on the target dataset (merrec). We utilize the Qwen2.5-7B model to generate high-quality text embeddings for all items. The RQ-VAE is then trained to quantize these embeddings into a sequence of discrete codes using a residual codebook structure.

  – **Potential Strength**: The learned vocabulary is perfectly aligned with the target domain's semantic distribution, as it is trained directly on the item descriptions.

  – **Critical Risk (Data Leakage)**: A naive implementation that trains the tokenizer on the full dataset would leak test-set information into the training process, invalidating downstream scaling law experiments.

- **Approach 3: In-Domain RQ-kmeans.** This method is similar to RQ-VAE but more stable and scalable. In our experiment we found using 6 layers of codebook and 256 codes per layer, the RQ-kmeans could reach collision rate 5.33% but RQ-VAE is stuck at 50.54%.

### B.2. Selected Methodology and Mitigation Protocol

We selected Approach 1 (SAE) as the our tokenization method after ablation against RQ-kmeans (6 layers of codebook and 256 codes per layer, after sweeping hyper-parameters Table 8). Figure 7 shows the comparison of scaling laws based on SAE vs RQ-Kmeans on 4B models. This figure shows SAE consistently outperformed RQ-Kmeans across all domains and steps.

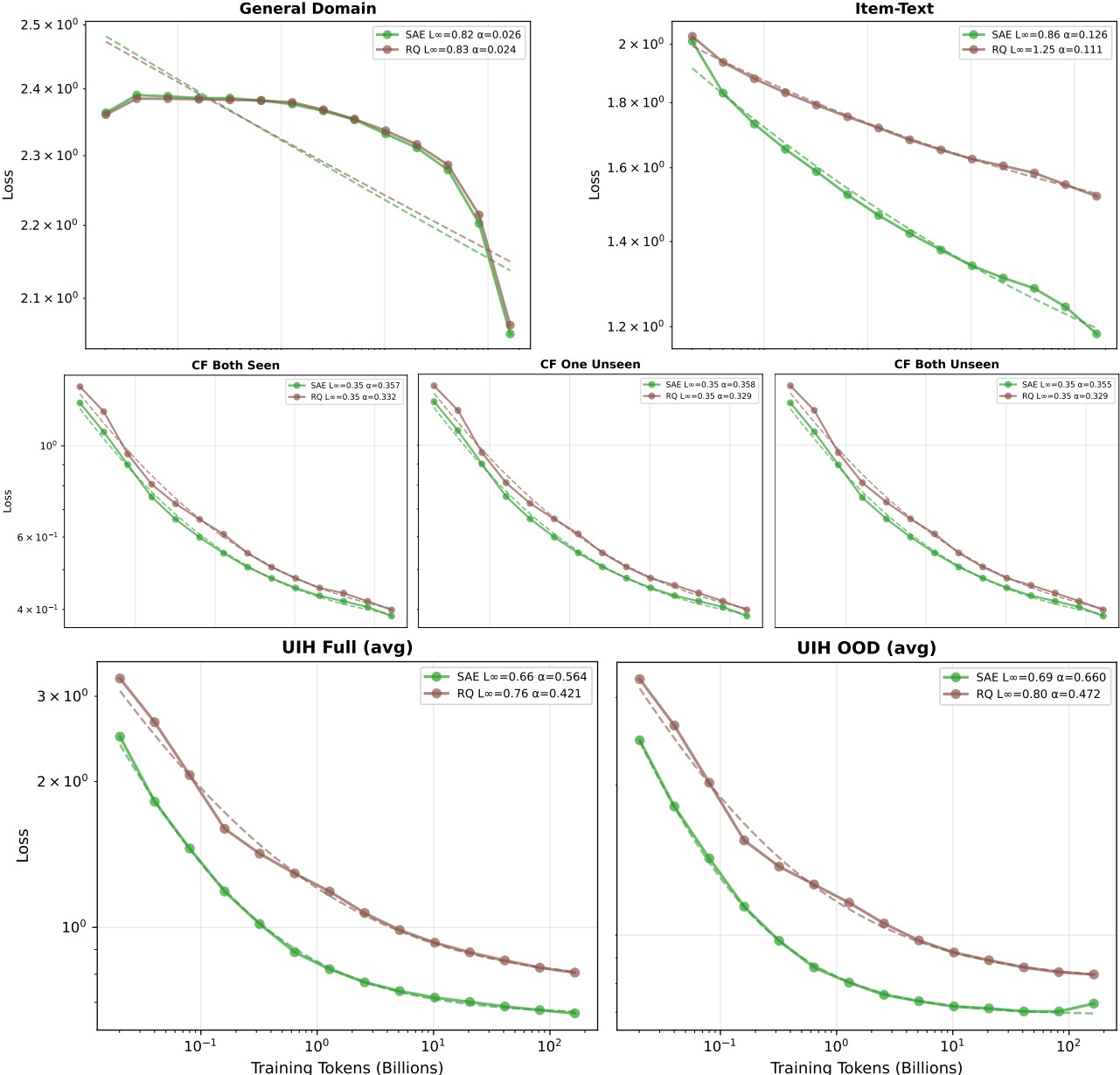

*Figure 7.* Scaling laws on 4B models with two tokenization methods SAE vs RQ-kmeans. The other settings are exactly the same. SAE clearly outperformed RQ-kmeans across all domains.

*Table 8.* RQ configuration sweep on Qwen-0.6B embeddings (64.1M items). All experiments use beam size 5 with progressive search and uniform sampling. Collision rate measured after uniform sampling.

| Layers | Codes/Layer | Total Codes | Collision % |
|--------|-------------|-------------|-------------|
| 3 | 256 | $256^3$ (16M) | 82.7 |
| 3 | 512 | $512^3$ (134M) | 37.4 |
| 5 | 256 | $256^5$ (1.1T) | 9.7 |
| 6 | 256 | $256^6$ (281T) | **3.5** |
| 8 | 128 | $128^8$ (72057T) | 1.2 |

## C. More Results on Ablation Study

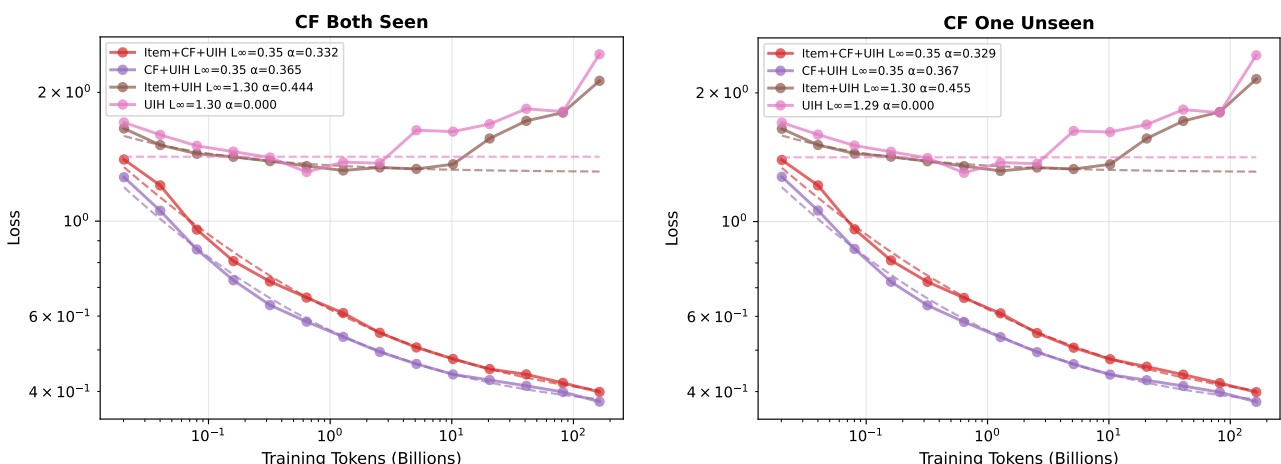

*Figure 8.* Results on CF Both Seen and CF On Unseen for ablation studies on selection of domains for recommendation data Figure 3. More results on ablation studies on data mixtures.

## D. Case Studies

In this section, we provide some examples of inferring the trained models. Figure 11 shows how the model recommend an item given the UIH.

Applying beam search in inference, the model could generate relevant and diverse responses, which is shown in Figure 12.

It would be interesting to find out how the model would react to a random UIH. We provide such example in Figure 13, which shows when given a random UIH, the model still tries to recommendation items relevant to items to UIH; but given the items in the UIH are random, the model's recommendations are more diverse.

Finally we provide an example of retrieving top-10 similar items given an generated item with semantic ID in Table 9. This example shows the effectiveness of the semantic ID in representing the items and capture their semantics.

## E. Quantitative Bias Audit

To rigorously characterize which biases our framework mitigates and which persist, we conducted a comprehensive hyperparameter sweep of our Node2Vec-based random walk algorithm across 14 configurations (100K sequences each). Table 10 summarizes the distributional properties of synthetic UIH compared to real user logs.

**Biases mitigated vs. retained.** Our framework eliminates two classes of bias by construction: (i) **positional bias**—the random walk has no concept of rank or presentation order; (ii) **temporal ordering bias**—each step depends solely on graph structure and $(p, q)$ parameters. However, **popularity bias** encoded in the co-occurrence graph edge weights is only partially mitigated: popular items naturally have higher-weight edges and thus higher visitation probabilities. The moderate Gini coefficients (0.635–0.732 vs. near-1.0 in real logs) indicate that our framework substantially reduces but does not eliminate

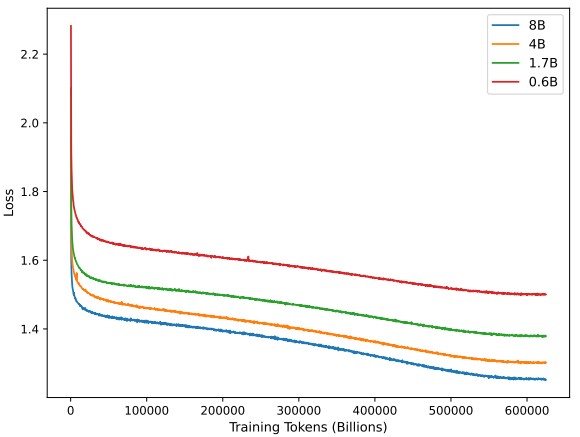 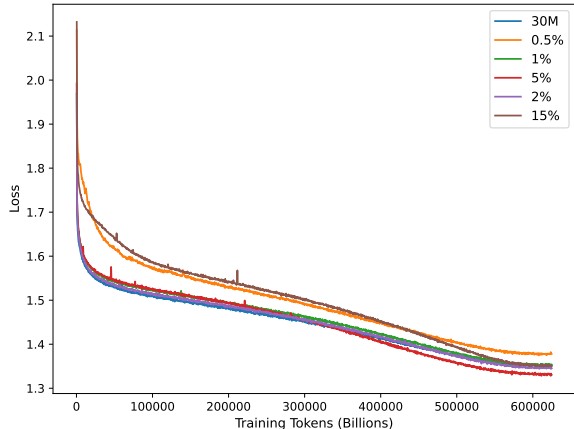

*Figure 9.* (a) The training loss curve for experiment in Figure 2. All the models are trained smoothly and show a monotonically decreasing training loss. (b) the training loss curve for experiment in Figure 5. The final training loss decreases in the order of $0.5\% > 1\% \approx 30M \approx 15\% > 2\% > 5\%$.

popularity concentration.

**Data isolation protocol.** We confirm full isolation across all experimental splits: (i) The CF graph is constructed exclusively from training-split interactions. Test items are either held out entirely (CF Both Unseen) or partially (CF One Unseen). (ii) The SAE tokenizer is trained on training-split item descriptions only. (iii) UIH evaluation sequences contain no overlap with training UIH sequences. (iv) No test-derived graph structure is used anywhere outside evaluation.

## F. Hyperparameter Sweep for Synthetic UIH Generation

We evaluated 14 distinct configurations for the Node2Vec-based random walk, including a 1st-order (DeepWalk-style) baseline and 12 variants of a 2nd-order, BFS-like strategy ($p$=0.5, $q$=2.0). The sweep varied $\alpha_{\text{stop}}$ (probability of terminating a walk at any step) and `path_conf_threshold` (minimum cumulative path confidence). Table 11 shows the full results; we selected the `1st_order_a015_thresh1e-09` configuration (highlighted), which achieves a good balance of coverage (180K items), low Gini (0.635), and high geometric lift (385.54).

The scaling law was found to be robust across the range of configurations tested. All 2nd-order walks used $p$=0.5 (moderate return probability) and $q$=2.0 (BFS-like exploration). Walk lengths ranged from 7.5 to 14.0 items depending on $\alpha_{\text{stop}}$ and threshold settings.

## G. Cross-Model and Cross-Dataset Generalization

This appendix provides additional details for the cross-generalization results summarized in the Discussion (Table 6).

**Model families.** We cover **Qwen3** (0.6B, 1.7B, 4B, 8B, 30B-A3B) and **Gemma 3** (1B, 4B, 12B)—two architecturally distinct families with different tokenizers and pretraining corpora. Both exhibit stable power-law scaling on a large-scale proprietary social media recommendation dataset with fundamentally different characteristics from MerRec (multimedia content, different interaction patterns, orders-of-magnitude larger catalog).

**Data scaling.** We trained Qwen3-4B with token budgets ranging from 216B to 1,104B tokens on the internal dataset. Scaling behavior remains predictable throughout, with no saturation in recommendation domains.

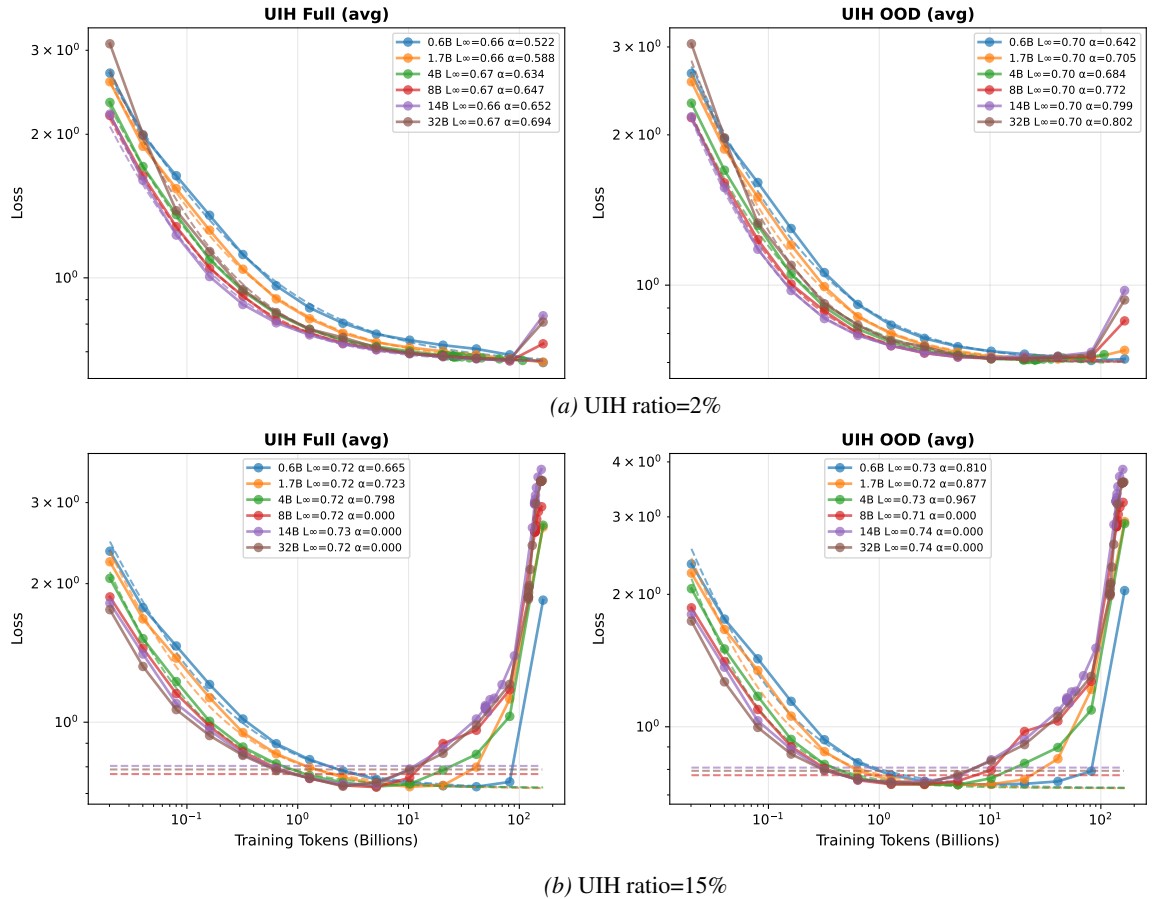

*(a)* UIH ratio=2%

*(b)* UIH ratio=15%

*Figure 10.* Scaling laws with mixture ratio on the reduced UIH data = (a) 2% and (b) 15% across different model scales. When the perplexity starts to increase depends on UIH mixture ratio, independent of model scales. We omitted the results on other domains as they shown the same patterns as Figure 5.

**Joint scaling law on the internal dataset.** The fitted joint scaling laws for Qwen3 on the internal dataset are:

$$\ell_{\text{general}} = 0.00 + 5.51\,N^{-0.262} + 1.51\,D^{-0.030} \tag{1}$$

$$\ell_{\text{item-text}} = 0.00 + 2.99\,N^{-0.169} + 0.73\,D^{-0.117} \tag{2}$$

$$\ell_{\text{UIH}} = 1.15 + 5.25\,N^{-0.380} + 6.72\,D^{-0.894} \tag{3}$$

The striking finding is that UIH on the internal dataset exhibits even stronger data scaling ($\beta$=0.894) than on MerRec ($\beta$=0.272), while the inverse relationship between model-size and data scaling exponents is preserved, reinforcing the conclusion that recommendation domains are fundamentally data-dominant.

**Case Study: Recommendation given UIH**

**Semantic ID Format**

**Prompt:**
```
A user interacted with the following
sequence of items:  <RECTOKEN> REC2981
REC2337 REC5067 REC1796 REC2985 REC6054
REC988 REC5454 </RECTOKEN>, <RECTOKEN>
REC2284 REC2581 REC2334 REC8032 REC6083
REC4502 REC5357 REC7604 </RECTOKEN>,
<RECTOKEN> REC5411 REC2334 REC6499
REC5622 REC2394 REC6054 REC7338 REC7704
</RECTOKEN>, <RECTOKEN> REC6594 REC2334
REC5261 REC4570 REC6576 REC906 REC6676
REC6499 </RECTOKEN>, <RECTOKEN> REC2284
REC641 REC8166 REC7604 REC2581 REC2312
REC7393 REC1231 </RECTOKEN>, <RECTOKEN>
REC5824 REC2334 REC4570 REC919 REC6513
REC2773 REC3722 REC7587 </RECTOKEN>,
<RECTOKEN> REC5824 REC7479 REC2334
REC5622 REC6499 REC344 REC3722 REC1553
</RECTOKEN>
```

**Response:** `REC5824 REC7479 REC2334 REC5622`
`REC2006 REC6499 REC1553 REC344`

**Ground Truth:**  `REC5824 REC7479 REC2334`
`REC344 REC5622 REC2006 REC6499 REC3722`

**Decoded Natural Language**

**Prompt:**
A user interacted with the following sequence of items: Alo top, by Alo in Women › Athletic apparel › Athletic Tank Tops. Size: XS (0-2)., Princess Polly Dress, by Princess Polly in Women › Dresses › Above knee, mini. Size: XS (0-2)., Bracelet, in Women › Jewelry › Bracelets., Necklace, in Women › Jewelry › Necklaces., Dress, in Women › Dresses › Knee-length. Size: S (4-6)., crossbody bag, in Women › Women's handbags › Crossbody Bags, Michael Kors purse, by Michael Kors in Women › Women's handbags › Shoulder Bags. Color: black.

**Response:** Michael Kors handbag, by Michael Kors in Women › Women's handbags › Shoulder Bags.

**Ground Truth:** Michael Kors purse, by Michael Kors in Women › Women's handbags › Shoulder Bags. Color: black/gold

*Figure 11.* An example of using the model to recommend item given the UIH. On the left, we provide the input and output with semantic ID format; and decoded natural language on the right. In this example, the model's response match perfectly with ground truth, except missing on the color.

**Case Study: Beam Search**

**Semantic ID Format**

**Prompt:**

```
A user interacted with the following
sequence of items:  <RECTOKEN> REC3972
REC7479 REC906 REC641 REC6621 REC4502
REC5992 REC2888 </RECTOKEN>, <RECTOKEN>
REC805 REC7607 REC2334 REC6499 REC1209
REC2633 REC6341 REC3072 </RECTOKEN>,
<RECTOKEN> REC3972 REC2590 REC2334
REC641 REC5454 REC5897 REC7479 REC4570
</RECTOKEN>, <RECTOKEN> REC2284 REC7604
REC641 REC2898 REC6581 REC2312 REC2581
REC748 </RECTOKEN>, <RECTOKEN> REC2284
REC641 REC7604 REC6581 REC2898 REC2581
REC8166 REC748 </RECTOKEN>, <RECTOKEN>
REC2284 REC641 REC2581 REC8166 REC748
REC2312 REC1231 REC7604 </RECTOKEN>,
<RECTOKEN> REC3972 REC6581 REC7479
REC2410 REC641 REC8174 REC3722 REC1231
</RECTOKEN>, <RECTOKEN> REC3375 REC2913
REC5824 REC7366 REC2752 REC984 REC1796
REC7604 </RECTOKEN>, <RECTOKEN> REC5824
REC82 REC5024 REC7427 REC1030 REC906
REC3968 REC2269 </RECTOKEN>, <RECTOKEN>
REC5824 REC2334 REC7479 REC1776 REC1553
REC2269 REC5622 REC6499 </RECTOKEN>
```

**Response   1:**    `REC5824 REC7479 REC2334`
`REC5024 REC82 REC6513 REC7736 REC1030`
**Response   2:**    `REC5824 REC7479 REC2334`
`REC5011 REC5622 REC2006 REC1553 REC3722`
**Response   3:**    `REC5824 REC7479 REC2334`
`REC5622 REC6499 REC344 REC3722 REC1553`

**Decoded Natural Language**

**Prompt:**

A user interacted with the following sequence of items: Vera Bradley Tote, by Vera Bradley in Women › Women's handbags › Tote Bags, Ring, in Women › Jewelry › Rings, Bath and Body Works, by Bath & Body Works in Beauty › Skin care › Body, Sol de Janeiro Brazilian Crush, by Sol de Janeiro in Beauty › Fragrance › Women, Marc Jacobs daisy, by MARC JACOBS in Beauty › Fragrance › Women, crossbody bag, in Women › Women's handbags › Crossbody Bags, Michael Kors Purse, by Michael Kors in Women › Women's handbags › Crossbody Bags. Color: other, Coach crossbody Medium purse, by Coach in Women › Women's handbags › Crossbody Bags, Tory Burch cross body bag, by Tory Burch in Women › Women's handbags › Crossbody Bags. Color: other, Michael Kors purse, by Michael Kors in Women › Women's handbags › Satchel.

**Response   1:**    `Coach Field tote, by Coach`
`in Women › Women's handbags › Shoulder`
`Bags.`
**Response 2:** `Michael Kors Purse, by Michael`
`Kors in Women › Women's handbags ›`
`Shoulder Bags.  Color:  brown.`
**Response 3:** `Michael Kors purse, by Michael`
`Kors in Women › Women's handbags ›`
`Shoulder Bags.  Color:  black.`

*Figure 12.* An example of applying beam search on the model's inference. On the left, we provide the input and output with semantic ID format; and decoded natural language on the right. In this example, the model's responses are relevant and diverse.

---

**Case Study: Recommendation given a Random UIH**

**Semantic ID Format**

**Prompt:**

A user interacted with the following sequence of items: <RECTOKEN> REC3311 REC953 REC418 REC7643 REC5624 REC419 REC7078 REC6875 </RECTOKEN>, <RECTOKEN> REC6812 REC6264 REC3311 REC7338 REC844 REC1746 REC237 REC5624 </RECTOKEN>, <RECTOKEN> REC6875 REC3311 REC2145 REC6576 REC5552 REC1796 REC2136 REC4685 </RECTOKEN>, <RECTOKEN> REC729 REC6261 REC4334 REC2044 REC1684 REC3927 REC4029 REC507 </RECTOKEN>, <RECTOKEN> REC5411 REC985 REC2394 REC7373 REC7338 REC4863 REC7366 REC6264 </RECTOKEN>, <RECTOKEN> REC7795 REC3245 REC856 REC6326 REC3812 REC6566 REC6855 REC7479 </RECTOKEN>, <RECTOKEN> REC3972 REC2145 REC7479 REC5454 REC5885 REC4540 REC6914 REC3812 </RECTOKEN>, <RECTOKEN> REC1166 REC237 REC984 REC1796 REC1295 REC5947 REC6341 REC6676 </RECTOKEN>, <RECTOKEN> REC6581 REC641 REC95 REC4647 REC5959 REC4208 REC4531 REC3722 </RECTOKEN>, <RECTOKEN> REC5695 REC4190 REC6261 REC2269 REC5624 REC1127 REC1796 REC729 </RECTOKEN>

**Response 1** REC4905 REC2334 REC1098 REC3339 REC6326 REC4190 REC2269 REC4863
**Response 2** REC641 REC2981 REC7479 REC524 REC6631 REC2334 REC2581 REC8032
**Response 3** REC641 REC2981 REC5980 REC4071 REC95 REC4531 REC2479 REC1685
**Response 4** REC641 REC2981 REC7479 REC524 REC6631 REC2334 REC573 REC3812
**Response 5** REC641 REC2981 REC7479 REC524 REC6631 REC2334 REC573 REC409

**Decoded Natural Language**

**Prompt:**

A user interacted with the following sequence of items: Play Air Maintenance Station Toy Plane, in Toys & Collectibles › Sports & Outdoor Play › Sand & Water Toys, Lego Star Wars Snowspeeder Pilot Light Gray Helmet, by LEGO in Toys & Collectibles › Building Toys › LEGO Toys, Thomas Wooden Railway Bill and Ben tank engine wood train 23102701, by Thomas & Friends in Toys & Collectibles › Remote Control Toys & Vehicles › Trains & Train Sets, Top Gun Flight Suit Mens Medium Large Costume Bradley Bradshaw Costume Halloween, by Top Gun in Men › Coats & jackets › Military. Size: M (38-40), Rainbow Eraser Multicolor Charm Handmade Friendship Bracelet, by Handmade in Women › Jewelry › Bracelets, <Fantastic four marvel visionaries John Byrne, by Marvel in Books › Fiction Books › Comics, Flia Shoes, in Kids › Girls shoes, samsung galaxy s8 spigen case. used, by Spigen in Electronics › Cell phones & accessories › Cases, covers & skins, Floral Cardigan by Talbots, by Talbots in Women › Sweaters › Cardigan. Size: M (8-10), VTG Sno Rider Womens Snow Pants Black 12 Insulated Bibs Snowmobile Zip up Legs, in Sports & outdoors › Apparel › Women. Size: L (12-14).

**Response 1** Vintage 90s Ford Mustang Overalls, by Ford in Men › Pants › Other. Size
**Response 2** Princess Polly, by Princess Polly in Women › Dresses › Above knee, mini. Size: M (
**Response 3** Women's size large tops bundle, by Boutique in Women › Tops & blouses › Blouse.
**Response 4** Dollskill x Alice in Wonderland, by Dolls Kill in Women › Tops & blouses › Other
**Response 5** Dolls Kill x The Powerpuff Girls – Bubbles & Butterflies Corset, by Doll

*Figure 13.* An example of model makes recommendation given a random UIH. On the left, we provide the input and output with semantic ID format; and decoded natural language on the right. In this example, the model's responses are still relevant to the items presented in the UIH.

*Table 9.* Example of running similarity search based on semantic IDs. Given the query (Row 1), we retrieval top-10 most similar items.

| Query | Similarity |
|---|---|
| REC5824 REC2334 REC4570 REC3722 REC7479 REC3968 REC2388 REC6054 Handbag, in Women › Women's handbags › Shoulder Bags | Query |
| REC5824 REC2334 REC4570 REC3722 REC7479 REC3968 REC2388 REC6054 Handbag, in Women › Women's handbags › Shoulder Bags | 100.0% |
| REC5824 REC2334 REC4570 REC3722 REC7479 REC3968 REC2388 REC2240 Beautiful Bag Designer Style!!, in Women › Women's handbags › Shoulder Bags | 87.5% |
| REC5824 REC2334 REC4570 REC3722 REC7479 REC3968 REC2388 REC7587 Light Gold Tote, in Women › Women's handbags › Shoulder Bags | 87.5% |
| REC5824 REC2334 REC4570 REC3722 REC7479 REC3968 REC2388 REC984 luxury bags, in Women › Women's handbags › Shoulder Bags | 87.5% |
| REC5824 REC2334 REC4570 REC3722 REC7479 REC3968 REC1030 REC2633 inspired bag, in Women › Women's handbags › Shoulder Bags | 75.0% |
| REC5824 REC2334 REC4570 REC3722 REC7479 REC3968 REC1228 REC5454 shoes women, in Women › Women's handbags › Crossbody Bags | 75.0% |
| REC5824 REC2334 REC4570 REC3722 REC7479 REC3968 REC1553 REC6842 Stay Real Bag, in Women › Women's handbags › Shoulder Bags | 75.0% |
| REC5824 REC2334 REC4570 REC3722 REC7479 REC3968 REC2240 REC1228 Fancy Handbag, in Women › Women's handbags › Shoulder Bags | 75.0% |
| REC5824 REC2334 REC4570 REC3722 REC7479 REC3968 REC2240 REC2388 beautiful bag, by Op in Women › Women's handbags › Shoulder Bags | 75.0% |
| REC5824 REC2334 REC4570 REC3722 REC7479 REC3968 REC2240 REC4075 Stylish Purse, Statement Piece, in Women › Women's handbags › Shoulder Bags | 75.0% |

*Table 10.* Bias audit comparing synthetic UIH against real user logs. Item Gini coefficient measures item exposure inequality (lower = more uniform). Synthetic UIH achieves substantially more uniform item exposure than real logs.

| Metric | Synthetic UIH | Real Logs |
|---|---|---|
| Item Gini coefficient | 0.635–0.732 (prod: 0.639) | >0.95 (top 1% items get 80% interactions) |
| Unique items per 100K seqs | 174K–189K | — |
| Semantic token coverage | 88.6–89.5% | — |
| Position bias | Zero by construction | CTR drops 50%+ from pos. 1 to 5 |
| Popularity bias | Partially mitigated (edge weights retain some popularity signal) | Strong feedback loops |

*Table 11.* Hyperparameter sweep for Node2Vec-based random walk UIH generation. Metrics computed on 100K sequences each.

| Config | $\alpha_{stop}$ | Thresh | Len | Items | Gini | Tokens | Geom Lift |
|---|---|---|---|---|---|---|---|
| bfs_a10_threshNone | 0.10 | None | 14.0 | 189K | 0.732 | 1,580 | 353.00 |
| bfs_a15_threshNone | 0.15 | None | 10.6 | 183K | 0.697 | 1,577 | 372.28 |
| bfs_a20_threshNone | 0.20 | None | 9.0 | 179K | 0.672 | 1,584 | 385.22 |
| bfs_a10_thresh1e-9 | 0.10 | 1e-9 | 8.6 | 178K | 0.660 | 1,574 | 368.91 |
| bfs_a15_thresh1e-9 | 0.15 | 1e-9 | 8.1 | 176K | 0.650 | 1,573 | 384.71 |
| bfs_a20_thresh1e-9 | 0.20 | 1e-9 | 7.6 | 174K | 0.643 | 1,568 | 388.84 |
| bfs_a10_thresh2e-9 | 0.10 | 2e-9 | 8.4 | 177K | 0.656 | 1,571 | 375.29 |
| bfs_a15_thresh2e-9 | 0.15 | 2e-9 | 8.0 | 175K | 0.648 | 1,569 | 384.51 |
| bfs_a20_thresh2e-9 | 0.20 | 2e-9 | 7.5 | 174K | 0.641 | 1,566 | 390.54 |
| bfs_a10_thresh3e-9 | 0.10 | 3e-9 | 8.3 | 177K | 0.655 | 1,569 | 376.11 |
| bfs_a15_thresh3e-9 | 0.15 | 3e-9 | 7.9 | 175K | 0.647 | 1,571 | 384.48 |
| bfs_a20_thresh3e-9 | 0.20 | 3e-9 | 7.5 | 174K | 0.639 | 1,570 | 392.59 |
| 1st_order_a015_threshNone | 0.15 | None | 10.6 | 188K | 0.680 | 1,579 | 376.80 |
| 1st_order_a015_thresh1e-9 | 0.15 | 1e-9 | 8.0 | 180K | 0.635 | 1,581 | 385.54 |

