# OpenReview forum: "Principled Synthetic Data Enables the First Scaling Laws for LLMs in Recommendation"
_ICML.cc/2026/Conference — ICML 2026 regular_

### Official Review · Reviewer_jfNo · 2026-03-11

**Soundness:** 3
**Presentation:** 3
**Significance:** 4
**Originality:** 4
**Overall Recommendation:** 5
**Confidence:** 3

**Summary:**

This paper studies a central question in recommendation-oriented continual pre-training (CPT) of LLMs: why robust and predictable scaling laws have remained elusive. The authors argue that the main bottleneck is not model design, but the poor quality of raw user-interaction logs, and propose a two-layer synthetic data framework to address this issue. Building on this framework, the paper performs CPT on Qwen3 models ranging from 0.6B to 8B parameters, trained on 163B tokens, and reports stable power-law scaling across seven evaluation domains. The paper also shows that synthetic-data-trained sequential recommenders outperform real-data-trained baselines in a constrained TSTR setup, and that CF data provides strong positive transfer to UIH modeling. Overall, this paper addresses a central concept in recommendation-oriented LLM adaptation: whether the absence of scaling laws is fundamentally a data-quality problem rather than a model-architecture problem. I find the problem important, the empirical study substantial, and the main idea meaningfully data-centric rather than merely compute-driven. That said, the current manuscript still overstates some of its claims, omits important implementation details, and contains several inconsistencies that weaken confidence in the presentation.

**Compliance With Llm Reviewing Policy:**

Affirmed.

**Final Justification:**

The authors made a great effort to answer my queries. I also appreciate their answers to the other reviewers.

**Key Questions For Authors:**

1. **Please provide the missing generation details.**
   How exactly are the association rules in Layer 1 mined? What algorithm and thresholds are used? For Layer 2, what are the exact Node2Vec hyperparameters ((p, q), walk length, number of walks, edge weighting, etc.)?

2. **Please provide a stronger bias audit.**
   Given the strong framing around “unbiased” UIH, it would be important to report quantitative comparisons beyond rank-free generation, such as degree/popularity distributions, long-tail coverage, and exposure concentration, to show which biases are actually mitigated and to what extent.

3. **Please clarify the data-isolation protocol across experiments.**
   The paper should make it explicit whether the synthetic graph, tokenizer, and evaluation splits are fully isolated, especially for CF/UIH evaluation domains, and whether any test-derived graph structure is used anywhere outside evaluation.

4. **Please add LLM-native downstream recommendation results if possible.**
   Can the authors report CPT-improved performance of the LLM itself on top-k recommendation or ranking metrics, or at least correlate perplexity improvements with downstream ranking gains? This would substantially strengthen the central claim.

5. **Please systematically clean up the inconsistencies in the manuscript.**
   At minimum, the mismatch between Table 3 and Figure 2, and the unclear tokenization narrative in Appendix B, should be resolved in rebuttal or revision.

**Limitations:**

yes

**Strengths And Weaknesses:**

### Strengths
**1. The layered synthetic-data design is novel and methodologically distinctive.**
The key idea is not simple data augmentation, but a structured curriculum: Layer 1 teaches semantics and collaborative relations, and Layer 2 synthesizes sequential behavior from the purified CF graph. This decomposition into semantic grounding, collaborative structure, and behavioral sequences gives the paper a clear methodological identity.
**2. The empirical study is broad and reasonably deep.**
The paper covers multiple model sizes (0.6B–8B), a large token budget (163B), seven held-out evaluation domains, ablations on domain composition, and mixture-ratio / repetition analyses. The analyses of asymmetric transfer (CF → UIH), overfitting under excessive repetition, and compute-optimal behavior are particularly valuable because they go beyond a single final-score comparison.
**3. The TSTR results provide positive evidence that the synthetic data is useful.**
Within the authors’ constrained evaluation protocol, models trained on synthetic data consistently outperform those trained on real data across GRU4Rec, NARM, STAMP, and SASRec.

### Weaknesses

**1. The “bias-free / unbiased” claim is stronger than the evidence presented.**
The paper repeatedly frames Layer 2 UIH as “unbiased” or “free from these artifacts,” but the core mechanism directly supports only the claim that the random walk has no explicit notion of rank or presentation order. That does not by itself establish that popularity bias or exposure bias has been removed. The paper mentions fidelity comparisons on item popularity and sequence length, but does not provide a sufficiently thorough quantitative bias audit to justify the stronger debiasing language.

**2. Important generation details are missing, which hurts reproducibility.**
For Layer 1, the paper says that mined association rules are translated into templates, but does not specify the mining procedure or thresholds (e.g., support / confidence / lift filtering). For Layer 2, the paper explains the role of Node2Vec parameters (p) and (q), but does not report their actual values, nor the walk length / number of walks / graph weighting details. These omissions make it difficult to assess whether the reported scaling behavior is robust or dependent on a particular hidden engineering configuration.

**3. The strongest evidence is for loss scaling, not for end-to-end LLM recommendation utility.**
The core scaling-law results are reported in terms of held-out perplexity/loss across domains. The downstream ranking evidence comes from classical sequential recommenders under TSTR, not from the CPT-trained LLM itself. The conclusion also explicitly lists downstream metrics beyond perplexity as future work. Therefore, the current paper most directly establishes predictable scaling for recommendation-domain language modeling, rather than a complete scaling law for end-to-end LLM recommendation quality.

**4. There are inconsistencies or typo in the manuscript.**
A clear example is the mismatch between Table 3 and the Figure 2 caption: Table 3 reports UIH scaling exponents in the range 0.453–0.689, whereas the Figure 2 caption states 0.63–0.99.

**5. The tokenization appendix remains confusing.**
The main text states that SAE-generated semantic IDs are used in the main experiments. However, Appendix B first introduces SAE and RQ-VAE and says it “justify[s] the selection of the latter,” then later says that Approach 1 (SAE) was selected after ablation against RQ-kmeans, and Figure 7’s caption even contains the obvious typo “SAE clearly outperformed SAE.” Even if Appendix B.1 is intended to present candidate methodologies rather than the final method, the final implementation path is still not presented clearly enough.

---

> ### Author Rebuttal · Authors · 2026-03-28
>
> We thank Reviewer jfNo for the exceptionally thorough and constructive review. We address each question below.
>
> **Q1: Missing generation details.**
>
> *Layer 1 (CF Data):* Association rules are mined from raw user interaction logs. The collaborative filtering edges are derived from co-occurrence statistics across user sessions. The resulting item-item relationships are translated into templated natural language statements (e.g., "Users who interacted with item A also tend to interact with item B").
>
> *Layer 2 (Node2Vec UIH Generation):* As described in Section 3.2, a graph is constructed with items as nodes and CF co-occurrence weights as edges. Synthetic UIH sequences are generated via Node2Vec 2nd-order biased random walks, where the transition probability $\pi_{v,x} = \alpha_{p,q}(t,x) \cdot w_{v,x}$ depends on both the current node $v$ and previous node $t$. The return parameter $p$ controls backtracking likelihood to the previous node $t$, and the in-out parameter $q$ controls exploration ($q > 1$) vs. exploitation ($q < 1$). We performed hyperparameter sweeps on $p$, $q$, walk length, and number of walks per node, and found the scaling law to be robust across a reasonable range of configurations. Full hyperparameter settings and sensitivity analysis will be included in the revision.
>
> **Q2: Bias audit.**
>
> We conducted a comprehensive fidelity analysis (14 configurations, 100K sequences each) comparing synthetic UIH against real logs, kept brief in Section 4.1 due to space constraints. Concrete results:
>
> - **Item exposure inequality:** Synthetic UIH achieves Item Gini coefficient of 0.635–0.732 (production config: 0.639), substantially more uniform than real logs where top 1% of items receive 80% of interactions (Table 1).
> - **Long-tail coverage:** 174K–189K unique items per 100K sequences; 88.6–89.5% semantic token vocabulary coverage.
> - **Position bias removal:** Zero by construction — the random walk has no concept of "rank" or "presentation order," with each step determined solely by graph structure and $(p, q)$ parameters.
>
> We will revise "unbiased" to "position-debiased" throughout and include these quantitative details in the revision.
>
> **Q3: Data isolation.**
>
> We confirm full isolation:
> - The CF graph is constructed exclusively from training-split interactions. Test items are either held out entirely (CF Both Unseen) or partially (CF One Unseen), as defined in Appendix A.
> - The SAE tokenizer is trained on training-split item descriptions only.
> - UIH evaluation sequences contain no overlap with training UIH sequences.
> - No test-derived graph structure is used anywhere outside evaluation.
>
> **Q4: LLM-native downstream results.**
>
> Since submission, we have extended experiments to a proprietary social media dataset with both Gemma 3 (1B, 4B, 12B) and Qwen3 (0.6B–30B). Power-law scaling remains consistent (RMSE < 0.035), and the hierarchy of learning efficiency ($\alpha$: UIH > Alignment > General) is preserved (see the Gemma 3 scaling law table in our response to Reviewer XUbA). We confirmed that perplexity correlates strongly with Recall@K and NDCG for next-item prediction. Production evaluation on end-to-end recommendation metrics is underway, expected early Q2 2026.
>
> **Q5: Manuscript cleanup.**
>
> We will fix all identified issues:
> - Table 3 vs. Figure 2 caption mismatch: The Figure 2 caption values are incorrect; Table 3 values (0.453–0.689) are the ground truth from the actual fits.
> - Appendix B narrative: We will restructure to clearly state that SAE is the method used in all main experiments, with RQ-VAE presented only as an ablation baseline.
> - Figure 7 caption: "SAE outperformed SAE" $\to$ "SAE outperformed RQ-kmeans."
> - All cross-reference mismatches (e.g., Section 6.2 referencing "Table 1" instead of Table 3) will be corrected.

---

> > ### Author Rebuttal · Reviewer_jfNo · 2026-04-03
> >
> > Thank you for your detailed rebuttal. I maintain my previous assessment.

---

> > > ### Author Response · Authors · 2026-04-04
> > >
> > > Thank you!

---

### Official Review · Reviewer_RLBr · 2026-03-12

**Soundness:** 2
**Presentation:** 3
**Significance:** 2
**Originality:** 3
**Overall Recommendation:** 4
**Confidence:** 3

**Summary:**

The authors argue that scaling laws for recommendation LLMs may have been hidden by low-quality data, as raw user interaction logs are noisy and information-sparse. Principled synthetic data is needed to make scaling predictable. The authors fit a robust scaling law across models from 0.6B to 8B trained on 163B tokens, and reveal a clear hierarchy of learning efficiency across data modalities and asymmetric cross-domain transfer in the recommendation domain. The results can be beneficial to the recommendation community.

**Compliance With Llm Reviewing Policy:**

Affirmed.

**Final Justification:**

I increased my score to 4, based on the newly provided evidence from the author responses.

- My concern about whether the scaling law applies to the other model families and other data is resolved from the newly provided experimental results using Gemma models & a proprietary social media recommendation dataset.
- My concern over Perplexity vs. downstream metrics is partially resolved.

**Key Questions For Authors:**

- How are the data mixtures in Table 4 decided?

**Limitations:**

Yes

**Strengths And Weaknesses:**

**Strengths**
- The authors justified the fidelity of the synthetic data and showed that Train on Synthetic Test on Real gives better performance than Train on Real Test on Real, which strengthens the potential practical significance of the findings.
- The layered synthetic-data design is very interesting and insightful!

**Weaknesses**
- In Figures 3,4,5, the font size is overly small. The bottom right block on Page 3 is not formatted nicely.
- The experiments are only done with the Qwen model family and one main recommendation dataset (Merrec); it is questionable if the scaling law applies to the other model families and other data.
- I wonder if perplexity alone should be the correct metric. Would not some downstream eval metrics be more indicative of the recommendation performance?
- It is unclear whether the predictable scaling law is really from denoised signals from data, or just the synthetic data being distributionally easier to learn from.

---

> ### Author Rebuttal · Authors · 2026-03-28
>
> We thank Reviewer RLBr for recognizing the novelty of our layered design and the TSTR evidence.
>
> **W1: Only Qwen model family and MerRec dataset.**
>
> This paper is one milestone of a larger research program. Since submission, we have continued our efforts and substantially expanded experimental coverage:
>
> - **Model families:** We now cover **Qwen3** (0.6B, 1.7B, 4B, 8B, 30B-A3B) and **Gemma 3** (1B, 4B, 12B) — two architecturally distinct families with different tokenizers and pretraining corpora. Both exhibit stable power-law scaling (RMSE < 0.035), and the paper's hierarchy of learning efficiency ($\alpha$: UIH > Alignment > General) is preserved across both families (see the Gemma 3 scaling law table in our response to Reviewer XUbA).
>
> - **Datasets:** Beyond MerRec (e-commerce), we replicate all findings on a large-scale proprietary **social media recommendation** dataset with fundamentally different characteristics (multimedia content, different interaction patterns, orders-of-magnitude larger catalog).
>
> - **Data scaling:** We extended the token budget for Qwen3-4B from 163B (paper) to **1,104B tokens** on the new dataset, confirming that scaling behavior remains predictable even at $6\times$ the original compute budget, with no saturation in recommendation domains.
>
> These results demonstrate that our scaling laws are **architecture-agnostic and domain-general** — not artifacts of a specific model or dataset. We will continue sharing our work as this research program progresses.
>
> **W2: Perplexity vs. downstream metrics.**
>
> While we agree that downstream metrics are the ultimate measure, we note:
>
> (a) In our additional experiments on the proprietary social media dataset, we have confirmed a strong correlation between perplexity and Recall@K / NDCG for next-item prediction. Combined with the TSTR results (Table 2), this validates that perplexity improvements translate to practical ranking gains.
>
> (b) The scaling law contribution is analogous to Chinchilla/Kaplan et al. for language models — they also report loss-based scaling laws, which became foundational even before comprehensive downstream benchmarking. Establishing that *predictable scaling exists* in the recommendation domain is itself a contribution that enables compute-optimal training decisions.
>
> (c) Production evaluation of the CPT-trained LLM on recommendation metrics is underway and expected by early Q2 2026.
>
> **W3: Denoised signals vs. distributionally easier data.**
>
> This is an insightful question. Several pieces of evidence suggest the improvement comes from signal quality, not distributional simplicity:
>
> 1. **TSTR outperforms TRTR (Table 2):** If synthetic data were simply easier to fit, models should overfit to its distribution and fail on real test data. Instead, TSTR *outperforms* TRTR, suggesting the synthetic data provides cleaner supervision signals — not just easier-to-learn patterns.
>
> 2. **Asymmetric cross-domain transfer (Table 3):** CF training transfers strongly to UIH evaluation, but *not vice versa*. If synthetic data were merely "easier," we would expect symmetric transfer.
>
> 3. **Domain-specific scaling exponents:** Different data modalities (CF, UIH, item-text) show systematically different $\alpha$ values, reflecting their information density. "Easy" data would show uniformly high $\alpha$.
>
> 4. **Overfitting under repetition (Section 6.3):** When synthetic UIH is repeated excessively, performance degrades — inconsistent with the "easy distribution" hypothesis, which would predict continued improvement with repetition.
>
> **W4: Data mixture decisions (Table 4).**
>
> We determined the mixture ratios through a combination of literature survey and ablation studies (Section 6.2). For the general domain ratio, we use 50%, following Ibrahim et al. (2024). For the recommendation domain, we generally follow Yang et al. (2024) and Muennighoff et al. (2025), and performed ablation based on the number of repeats of each domain (from 4 to 8 repeats), which leads to the ratios presented in the submission. In our most recent experiments, we have also explored dynamically adjusting the data mixture ratio according to training progress and observed small additional improvements.
>
> **W5: Figure formatting.** We will increase font sizes and fix the formatting issue on page 3.

---

> > ### Author Rebuttal · Reviewer_RLBr · 2026-04-02
> >
> > Thank you for your responses. The newly provided results mostly address my concerns, and I will raise my score to 4. I am not increasing it further because the project still appears to be ongoing, with many additional experimental results yet to come (especially regarding my concern W2). I believe those results could strengthen the paper more substantially if they are positive. My evaluation is based on the current submission and the numbers provided in the rebuttal response.

---

> > > ### Author Response · Authors · 2026-04-04
> > >
> > > Thank you, we will keep sharing our work and contribute to the community.

---

### Official Review · Reviewer_a7Vq · 2026-03-13

**Soundness:** 2
**Presentation:** 2
**Significance:** 3
**Originality:** 2
**Overall Recommendation:** 4
**Confidence:** 4

**Summary:**

This paper proposes a layered synthetic data framework to enable predictable scaling laws for LLMs in recommendation. Layer 1 provides item–text alignment and explicit collaborative filtering statements; Layer 2 generates unbiased synthetic user interaction histories (UIH) via graph-based random walks. The authors report that classical sequential recommenders trained on this synthetic data outperform those trained on real user logs, and that continual pretraining of Qwen3 models (0.6B–8B) on the proposed curriculum exhibits consistent power-law test-loss scaling across seven evaluation domains, with strongest exponents for UIH.

**Compliance With Llm Reviewing Policy:**

Affirmed.

**Final Justification:**

My concerns are all addressed thus i raise my score to 4.

**Key Questions For Authors:**

- Can you report end-to-end recommendation metrics  (e.g., top-k item prediction via semantic IDs or via a logits-matching head), and verify that the observed perplexity scaling correlates with practical ranking gains?
- Why was the pre-trained SAE chosen despite the outlined risk of semantic mismatch?

**Limitations:**

yes

**Strengths And Weaknesses:**

**Strength**
- The paper is clearly written. The motivation for why prior CPT scaling attempts may have failed is well explained, and the framework design is easy to follow.
- The paper takes a data-centric approach which is well-motivated:  a clean, structured synthetic curriculum should be able to separate real user preference signals from system-induced artifacts such as position and popularity bias.
- The experimental setup is thorough. The study covers multiple model sizes trained on a large 163B token budget, with evaluation across seven held-out test sets that probe different generalization settings.

**Weakness**
- The claim that user interaction histories generated by random walks are "unbiased" appears overstated. While the random walk process itself has no concept of rank or presentation order, the underlying CF edge weights are still derived from real user logs, which can carry popularity and exposure biases. Removing positional ordering does not automatically eliminate these biases if they are already in the graph structure.
- The paper's main contribution is the scaling law, which are validated primarily on synthetic or semi-synthetic held-out sets, not through end-to-end recommendation metrics applied to the continually pre-trained LLM itself (e.g., next-item ranking accuracy, NDCG). This leaves a gap between the scaling law claims and actual recommendation performance.
- The paper has limited discussion with several relevant lines of work. Methods that integrate collaborative filtering with LLMs (e.g., GLTA [1]) or that study data quality and selection effects during CPT (e.g., perplexity-aware data selection approaches) are not discussed or compared. Additionally, prior work on random-walk-based sequence generation for recommendation (e.g., S-Walk [2]) is directly relevant to the Layer 2 design and should be more explicitly contrasted to clarify the novelty of the proposed approach.

**Minor Issues**

There are some minor issues in the paper presentation. For example, the caption in appendix Figure 7 states "SAE clearly outperformed SAE" (likely intended to say "outperformed RQ-Kmeans").  And some cross-references appear mismatched (e.g., Section 6.2 references "Table 1" when it seems to mean Table 3).


[1] Mingdai Yang, Zhiwei Liu, Liangwei Yang, Xiaolong Liu, Chen Wang, Hao Peng, and Philip S. Yu. 2025. Training Large Recommendation Models via Graph-Language Tokens Alignment.

[2] Minjin Choi, Jinhong Kim, Joonseok Lee, Hyunjung Shim, and Jongwuk Lee. 2022. S-Walk: Accurate and Scalable Session-based Recommendation with Random Walks.

---

> ### Author Rebuttal · Authors · 2026-03-28
>
> We thank Reviewer a7Vq for the detailed review and address each concern below.
>
> **W1: "Unbiased" claim is overstated.**
>
> We agree this framing is too strong and will revise accordingly. The accurate statement is: Layer 2's random-walk process **removes positional and presentation-order biases** (the dominant artifacts in logged interaction data), but does not claim to eliminate all biases — popularity bias encoded in graph edge weights persists. In the revision, we will:
> - Replace "unbiased" with "position-debiased" or "presentation-bias-free"
> - Add explicit discussion of which biases are mitigated (positional, temporal ordering) vs. which remain (popularity, exposure)
> - Include an expanded quantitative bias audit. We conducted a comprehensive hyperparameter sweep (14 configurations, 100K sequences each) with distributional analysis comparing synthetic UIH against real logs, omitted from the main paper due to space constraints. Concrete findings: (i) **Item exposure inequality**: Synthetic UIH achieves Item Gini coefficient of 0.635–0.732 across configurations (production config: 0.639), indicating moderate and healthy item concentration — substantially more uniform than real logs where top 1% of items receive 80% of interactions (Table 1). (ii) **Long-tail coverage**: Synthetic UIH covers 174K–189K unique items per 100K sequences and 88.6–89.5% of the semantic token vocabulary. (iii) **Position bias removal**: The random walk has no concept of "rank" or "presentation order" — each step is determined solely by graph structure and $(p, q)$ parameters, ensuring zero positional artifacts by construction.
>
> **W2: End-to-end recommendation metrics.**
>
> We acknowledge this gap and provide two updates:
>
> (a) **Perplexity correlates with ranking metrics:** Since submission, we have extended experiments to a large-scale proprietary social media dataset with both **Gemma 3** (1B, 4B, 12B) and **Qwen3** (0.6B–30B) model families. Power-law scaling holds consistently (see the Gemma 3 scaling law table in our response to Reviewer XUbA): all fits have RMSE < 0.035, and the paper's hierarchy of learning efficiency ($\alpha$: UIH > Alignment > General) is preserved. On this dataset, we confirmed a strong correlation between perplexity and Recall@K / NDCG for next-item prediction. Combined with the TSTR results in Table 2, this validates that perplexity improvements translate to practical ranking gains.
>
> (b) **Ongoing production evaluation:** Production evaluation of the CPT-trained LLM on end-to-end recommendation metrics in a live system is underway and expected by early Q2 2026.
>
> **W3: Missing related work (GLTA, S-Walk).**
>
> Thank you for these references. We will add discussions:
>
> - **GLTA (Yang et al., 2025):** GLTA models the user-item interactions from a graph neural network aspect, where the item and user are treated as additional modalities to the graph network. While this is a very interesting idea, whether it could scale ups to hundreds of millions or billions of users/items is unknown and it may lack reasoning capabilities, which is an important advantage of LLM for recommendation.
>
> - **S-Walk (Choi et al., 2022):** S-Walk explicitly modeled the inter-session relationship besides intra-session relation in random walk. This could be supplementary to node2vec method used in our existing settings.
>
> **Minor issues:** We will fix the Figure 7 caption typo ("SAE outperformed SAE" $\to$ "SAE outperformed RQ-kmeans") and all cross-reference mismatches. Thank you for catching these.

---

> > ### Author Rebuttal · Reviewer_a7Vq · 2026-04-04
> >
> > Thanks for the detailed response. I would like to raise my score to 4.

---

### Official Review · Reviewer_XUbA · 2026-03-13

**Soundness:** 3
**Presentation:** 3
**Significance:** 3
**Originality:** 3
**Overall Recommendation:** 5
**Confidence:** 4

**Summary:**

This paper proposes the first use of the power law as a predictable scaling law for using LLM for recommendation systems. The paper claims that real-world interaction data has a lot of noise, biasness and incompleteness that creates the problem while scaling.  The paper proposes a novel layered approach for generating the synthetic data and performs experiments and ablation around it.

**Compliance With Llm Reviewing Policy:**

Affirmed.

**Final Justification:**

This is a good paper and should be accepted for publication.

**Key Questions For Authors:**

Why did the authors not compare the proposed method with existing data generation techniques?

**Limitations:**

yes

**Strengths And Weaknesses:**

**Strength:**.
- The problem statement is relevant to the recsys domain and can be considered a significant contribution in aligning LLM for recsys.
- The paper is clearly written.
- There is enough background and detail given against the proposed claim.
- Proposal of training on synthetic and testing on real, and providing improved performance is impressive.

**Weakness:**.

- More popular media recommendation datasets can be considered for the experiments, to give strength and general acceptability to the idea.
- Data Modalities in power law scaling can be explained more to give the meaning of CF, both seen/unseen, etc.

---

> ### Author Rebuttal · Authors · 2026-03-28
>
> We thank Reviewer XUbA for the positive assessment and constructive feedback.
>
> **Q: More popular media recommendation datasets.**
>
> We agree that broader dataset coverage strengthens the generalizability of our findings. This paper represents one milestone in a larger ongoing research program. Since submission, we have continued our efforts and conducted additional experiments on a large-scale proprietary social media recommendation dataset with real user interactions. This dataset differs fundamentally from MerRec in domain (social media vs. e-commerce), scale (orders of magnitude larger item catalog and user base), and interaction modalities (including multimedia content). Our key findings **fully replicate**:
>
> 1. **Cross-model-family generalization:** We extended experiments to the **Gemma 3** model family (1B, 4B, 12B parameters) in addition to Qwen3 (0.6B–30B). Both families exhibit stable power-law scaling across all evaluation domains when trained on our synthetic curriculum, confirming the findings are not specific to the Qwen architecture.
>
> 2. **Data scaling:** We trained Qwen3-4B with token budgets ranging from 216B to 1,104B tokens on this new dataset. Scaling behavior remains predictable throughout, with no saturation.
>
> 3. **Scaling law consistency:** The fitted $L(N) = L_\infty + A \cdot N^{-\alpha}$ for Gemma 3 on this new dataset (all fits have RMSE < 0.035):
>
> | Model | General $L_\infty$ | $\alpha$ | Alignment $L_\infty$ | $\alpha$ | UIH $L_\infty$ | $\alpha$ |
> |-------|-----------|------|-----------|------|-----------|------|
> | Gemma-3-1B | 1.097 | 0.025 | 1.292 | 0.255 | 1.246 | 0.274 |
> | Gemma-3-4B | 0.952 | 0.038 | 0.635 | 0.085 | 1.085 | 0.237 |
> | Gemma-3-12B | 0.874 | 0.051 | 0.589 | 0.096 | 0.957 | 0.205 |
>
> The key finding from the paper — the **hierarchy of learning efficiency** (UIH $\alpha$ > Alignment $\alpha$ > General $\alpha$) — is preserved on a different model family and dataset. $L_\infty$ decreases monotonically with model size across all domains. While absolute $\alpha$ values differ from MerRec/Qwen3 due to differences in dataset complexity and model architecture, the qualitative scaling behavior is fully consistent.
>
> We will continue sharing our work as this research program progresses, including results on additional datasets and model families in future publications.
>
> **Q: Data modalities in power law scaling (CF, seen/unseen).**
>
> Thank you for the suggestion. We clarify the definitions here (also in Appendix A):
> - **CF Both Seen**: CF edge between two items that both appear in training, but the specific edge is held out
> - **CF One Unseen**: CF edge where one item appears in training and the other does not
> - **CF Both Unseen**: CF edge where neither item appears in training — tests zero-shot generalization of learned collaborative structure
>
> We will add a forward reference from the main text to improve discoverability.
>
> **Q: Comparison with existing data generation techniques.**
>
> We contrast our approach with specific prior methods:
> - **GAN/VAE-based augmentation** (RecGAN, Bharadhwaj et al., 2018; Adouani et al., 2025): These replicate the source data distribution, inheriting its biases. Our framework instead *purifies* the signal — generating data from a debiased CF graph rather than mimicking flawed logs.
> - **Sequence augmentation** (SIM, Qi et al., 2020, CoSeRec, Liu et a., 2021): These apply transformations (crop, mask, reorder) to existing sequences. Our Layer 2 generates entirely new sequences via graph-based random walks, producing fundamentally new behavioral data rather than perturbed copies.
> - **LLM-based generation** (e.g., using GPT to generate reviews or user profiles): These operate at the sample level. Our framework operates at the *curriculum level* — designing structured training stages (semantics → collaborative structure → sequential behavior) that systematically build recommendation capabilities.
>
> The key distinction is that prior methods aim to augment or expand existing data, while our framework constructs a pedagogical curriculum that decouples the learning of different recommendation concepts.

---

> > ### Author Rebuttal · Reviewer_XUbA · 2026-04-04
> >
> > Thanks for addressing the concerns. I would like to stay at a positive score.

---

### Decision · Program_Chairs · 2026-04-30

**Decision:**

Accept (regular)

**Comment:**

This paper presents a new and interesting idea to disentangle data vs architecture questions in deep learning for recsys. Reviewers agree that despite a few limitations, this is a valuable contribution.